**communications** engineering

# An Integrative lifecycle design approach based on carbon intensity for renewable-battery-consumer energy systems
Aoye Song [1,2] & Yuekuan Zhou [1,2,3,4] ✉

Driven by sustainable development goals and carbon neutrality worldwide, demands for both renewable energy and storage systems are constantly increasing. However, the lack of an appropriate approach without considering renewable intermittence and demand stochasticity will lead to capacity oversizing or undersizing. In this study, an optimal design approach is proposed for integrated photovoltaic-battery-consumer energy systems in the form of a $m^2$-kWp-kWh relationship in both centralized and distributed formats. Superiorities of the proposed matching degree approach are compared with the traditional uniformity approach, in photovoltaic capacity, battery capacity, net present value and lifecycle carbon intensity. Results showed that the proposed method is superior to the traditional approach with higher net present value and lower carbon intensity. Furthermore, the proposed method can be scaled and applied to guide the design of photovoltaic-battery-consumer energy systems in different climate zones, promoting sustainable development and carbon neutrality globally.

Faced with the energy shortage crisis, deteriorated environmental issues, and global warming in 21 centuries, along with the Paris Agreement, countries worldwide are imposing sustainability development and controlling carbon emissions at an ever-high level. For example, China has announced that it will achieve a carbon peak by 2030 and carbon neutrality by 2060[1]. As an essential part of the carbon neutrality plan in carbon neutrality transformation, the power grid with traditional fossil fuel power plants requires energy structure recombination and reformation with large renewable penetrations[2,3] and battery storages[4,5]. Although the renewable energy-driven operational stage is clean, these devices (especially for battery storage equipment[6]) are carbon-intensive[7,8] due to a great amount of embodied carbon in their manufacturing, operation, and recycling processes[9]. Therefore, optimizing battery capacity is essential to avoid material waste, and economic-environmental infeasibility caused by undersized or oversized battery capacity.

Currently, the common battery capacity optimization methods are generally based on economic indicators[10,11], technical indicators[12], or a combination of these two indicators[13]. Economic indicators consider capital costs, daily O&M (operation and maintenance) costs, and fuel costs, with the aim of pursuing the highest net present value and the lowest levelized electricity costs[14]. However, as dependent indicators, net present value or

levelized electricity cost is highly dependent on independent variables (e.g., renewable cost[15], battery prices[16] and electricity prices[17]) or intermediate transition variables (e.g., surplus renewables, load shortage, battery charge and discharge powers, battery degradation rates[18]), failing to reveal basic and fundamental physical mechanisms on dynamic interconnection and power interaction within the integrated photovoltaic-battery-consumer energy systems[13]. Therefore, economic indicators can be used for technology comparison and to assist decision-making, but not suitable for capacity sizing. Meanwhile, technical indicators usually consider the battery's voltage and frequency regulation capabilities[13], but there is a lack of technical indicators to optimize battery capacity based on the inherent relationship between PV (photovoltaic) generation and building energy consumption. Since achieving carbon neutrality requires at least 80% renewable energy share[19], increasing the penetration rate of renewable energy is crucial to achieving carbon neutrality. In addition, the problem of PV generation ageing and battery capacity degradation will also cause uncertainty in battery capacity sizing[18]. When selecting the sizing of a solar PV-battery system, references[20–22] consider both the generation and residential sides along with the corresponding technical and economic indicators. However, it lacks consideration for the dynamic degradation of PV generation and battery capacity, which can cause major errors in optimizing the system size. Up

[1]Sustainable Energy and Environment Thrust, Function Hub, The Hong Kong University of Science and Technology (Guangzhou), Nansha, Guangzhou, 511400 Guangdong, China. [2]Division of Emerging Interdisciplinary Areas, The Hong Kong University of Science and Technology, Clear Water Bay, Hong Kong SAR, China. [3]Department of Mechanical and Aerospace Engineering, The Hong Kong University of Science and Technology, Clear Water Bay, Hong Kong SAR, China. [4]HKUST Shenzhen-Hong Kong Collaborative Innovation Research Institute, Futian Shenzhen, China. ✉e-mail: yuekuan.zhou@outlook.com

until now, there is a lack of appropriate method for sizing battery capacity in distributed energy systems, and inappropriate methods can result in the undersizing or oversizing of battery capacity, material waste and economic-environmental infeasibility.

Furthermore, the deployment of renewable energy in the power grid also involves the choice between centralized and distributed systems[23]. Centralized systems can achieve better large-scale renewable energy development, improve energy utilization efficiency, and reduce the volatility and uncertainty of the power grid[24,25]. By contrast, distributed systems can better utilize dispersed renewable energy resources, reduce transmission losses and construction costs, and flexibly meet personalized user needs[26]. However, optimizing battery capacity in distributed systems presents huge challenges. Unlike centralized PV-battery-consumer systems that focus primarily on intermittent renewable energy charging, batteries in distributed prosumer-battery systems will simultaneously balance on-site renewable energy supply and energy demand[27], which poses a huge challenge in optimizing battery capacity sizing. In addition, considering the overlap in operational stages and separation in other stages of each integrated component, lifecycle carbon emissions and carbon intensity of an integrated building-photovoltaic-battery system have not been quantified[28]. Furthermore, the impact of capacity sizing methodology on lifecycle carbon footprint has not been analysed, comprehensively considering the embodied carbon emission in raw materials of renewable systems and batteries, as well as the operational carbon emission with their dynamic performance degradation and replacement times[29]. In addition, in terms of electrified lifecycle sustainable transformation, either a centralized or distributed energy system is the most effective solution is still questionable. Comparison between centralized and distributed energy systems has not been systematically conducted, in terms of lifecycle carbon intensity and net present value. Guidelines on future battery circular economy with sustainability are not clear in planning, designing and operating.

Therefore, research gaps can be summarized as follows:

1. Considering the carbon intensity of batteries in renewable systems, optimizing battery capacity is crucial to avoiding material waste and economic infeasibility. However, current methods for battery capacity sizing have disadvantages (e.g., most are based on economic indicators and do not reveal the fundamental physical mechanisms of dynamic interconnection and power interaction within integrated photovoltaic-battery-consumer energy systems), leading to capacity overestimation or underestimation. A few methods based on technical indicators only apply to centralized systems or PV/wind turbine farms, and are not suitable for distributed systems.

2. Centralized photovoltaic systems focus primarily on providing consistent electricity, while distributed consumer battery systems face the challenge of balancing both on-site renewable energy supply and energy demand simultaneously, making it difficult to optimize battery capacity. In addition, the dynamic degradation of battery capacity and PV power generation will also cause difficulties in battery capacity optimization.

3. The quantification of lifecycle carbon emissions and carbon intensity for building-photovoltaic-battery integrated systems is complicated due to incomplete database on embodied carbon emission, process uncertainty and overlap of each integrated component's operating stage. Additionally, there has been no comparison of centralized and distributed energy systems in terms of lifecycle carbon intensity and net present value for sustainable electrification lifecycle transformation. As a result, clear sustainable development guidelines for the future battery circular economy are still needed for planning, designing, and operation.

Based on above research gaps, research originality and contributions are listed below:

1. Developing a scientific and generic methodology for capacity sizing under the distributed prosumer-battery framework considering demand-side management, PV generation degradation and battery dynamic ageing, which will ensure the techno-economic-environmental performance optimality and avoid capacity undersizing or oversizing and resource waste.

2. Quantifying the lifecycle carbon intensity of an integrated building-photovoltaic-battery system with an overlap in operation stages, together with performance superiority through comparison between centralized and distributed energy systems in lifecycle carbon intensity and net present value.

3. Exploring life cycle assessment (LCA) and zero-carbon transformation of both centralized PV-battery-consumer and distributed prosumer-battery systems in different climate zones, and developing climate-adaptive zero-carbon circular economy with a systematic battery capacity sizing methodology and clear guidelines for sustainability transition in planning, designing and operating.

Therefore, this study includes five aspects: 1) establishing centralized PV-battery-consumer and distributed prosumer-battery energy systems; 2) proposing a scientific method for quantifying battery storage capacity while considering embodied and operational carbon emission in renewable systems and batteries with their dynamic performance degradation and replacement times; 3) quantifying lifecycle carbon emissions and carbon intensity of the integrated PV-battery-consumer system; 4) proposing design guidelines for the PV-battery-consumer energy system; 5) proposing zero-carbon design guideline with LCA analysis of integrated PV-battery-consumer energy systems with climate adaptability.

## Results and discussions

### The lifecycle design for the renewable-battery-prosumer system

Figure 1 illustrates the model development, system establishment, and circular economy framework of this study. As shown in Fig. 1a, the basic information required for this study is shown in a schematic form, including meteorological parameters, electricity price data, renewable energy distribution, and power sources of the grid. Figure 1b reflects the system design of this study, which includes the following three systems:

Traditional energy systems do not install any PV panels, but instead rely on thermal power generation for electricity supply, resulting in a high electricity carbon emission factor (about $0.8 \sim 1.0$ kg $CO_2$e kWh$^{-1}$ in different regions)[7]. In addition, there exists an imbalance between power generation and demand in traditional systems, which is addressed by regulating the frequency of thermal power generators to achieve power control. With the gradual transition from traditional energy systems to renewable energy systems, two potential future energy systems have emerged: centralized PV-battery-consumer systems and distributed prosumer-battery systems.

Centralized PV-battery-consumer systems use solar energy to cover the building energy demand. However, the power generation characteristics of PVs are not as stable as traditional thermal power plants, and they fluctuate greatly over time. Therefore, the design idea of centralized PV-battery-consumer energy systems is to concentrate all the energy generated by PVs and supply electricity in a similar way to traditional thermal power plants by energy storage systems. The limitation of this approach is that the energy storage is not set up with building energy demand, which will cause the over- or under-estimation of the battery capacity.

In distributed prosumer-battery systems, the design idea is that it treats each building-PV-battery energy system as an independent system and can still interact with the external power grid as a whole system. Therefore, this system can fully and flexibly consider the spatiotemporal mismatching between solar energy and building demand when designing the energy storage system, and better configure the required energy storage battery capacity.

Figure 1c illustrates the design approach for the PV and battery capacity in the system. In traditional systems, electricity generation can be adjusted by regulating the generator frequency, but this is not possible in renewable energy systems. Therefore, in this study, a centralized renewable energy system is developed to supply stable power through energy storage

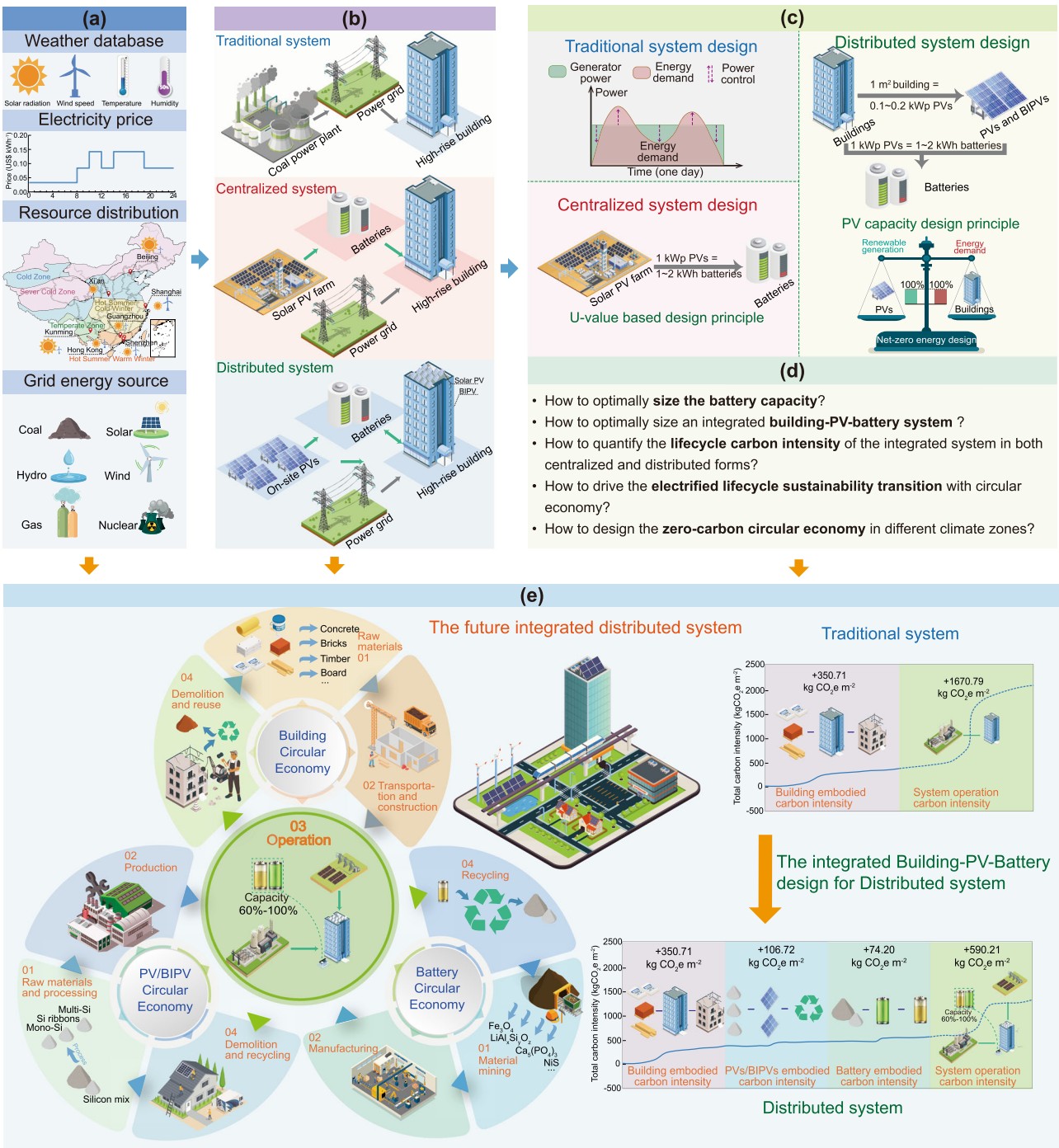

**Fig. 1 | Lifecycle design and approach for energy systems. a** Basic information for the system, including weather data, electricity price, renewable resource distribution and grid power source. **b** Designs of traditional energy systems, centralized PV-battery-consumer systems and distributed prosumer-battery systems. **c** Optimal system sizing approaches for centralized and distributed system. **d** research questions of the study. **e** Lifecycle assessment and design guidelines for integrative energy systems. (Note: PV and BIPV refer to photovoltaic and building-integrated photovoltaic.)

systems, and the battery capacity is optimized using the U-value method[26]. In distributed energy systems, after determining the required PV capacity based on the net-zero energy principle, battery capacity optimization is achieved by using the M-value method developed in this study. Figure 1d shows the five most important research questions addressed in this study, while Fig. 1e further illustrates the application of the circular economy framework in the building energy system. The building lifecycle is categorized into four stages: raw materials, transportation and construction, operation, and demolition and reuse. Likewise, the photovoltaic lifecycle is classified into four stages: raw materials and processing, production,

operation, demolition and recycling. Similarly, the battery's lifecycle is divided into four stages: materials mining, manufacturing, operation, and recycling. It is important to note that these stages overlap during the operation stage. Therefore, it is necessary to differentiate between embodied carbon and operational carbon for the LCA of the entire energy system. Furthermore, the lifecycle carbon intensity chart in Fig. 1e shows the carbon emission changes from traditional systems to distributed energy systems for a hotel building. The total carbon intensity of the traditional system is as high as 2021.50 kg $CO_2$e m$^{-2}$ (including 350.71 kg $CO_2$e m$^{-2}$ of building embodied carbon intensity and 1670.79 kg $CO_2$e m$^{-2}$ of operational carbon

**Fig. 2 | The battery capacity sizing methods.**
**a** U-value (Uniformity value) calculation for centralized PV (photovoltaic)-battery-consumer system. **b** M-value (Matching value) calculation for distributed PV-battery-building system. **c** battery capacity by U-value approach in the centralized PV-battery-consumer system. **d** battery capacity by M-value approach in the distributed PV-battery-building system.

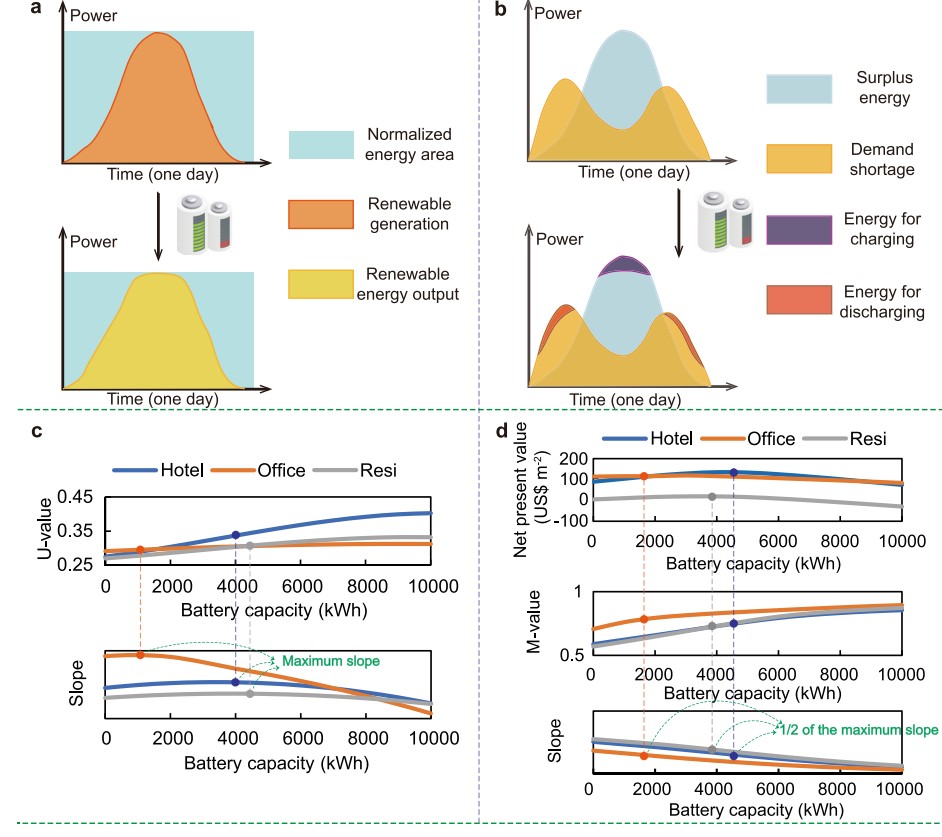

intensity). After designing the system by the distributed building-PV-prosumer guidelines proposed in this study, the total carbon intensity of the system decreased to 1121.83 kg $CO_2$e m$^{-2}$. While the embodied carbon intensities of the building, PVs, and batteries are 350.71, 106.72, and 74.20 kg $CO_2$e m$^{-2}$, respectively, and the system's operational carbon emissions are 590.21 kg $CO_2$e m$^{-2}$. This demonstrates the effectiveness of the distributed building-PV-prosumer design guidelines proposed in this paper.

### Battery capacity sizing—centralized PV-battery-consumer system and distributed prosumer-battery system
Centralized PV-battery-consumer system: Capacity sizing on centralized PV-battery-consumer systems includes two aspects, i.e., the PV capacity sizing and battery capacity sizing. In terms of centralized PV-battery-consumer systems, the PV capacity sizing is based on the zero-energy principle, i.e., the electricity production of PVs needs to meet the building energy consumption in total. However, the battery capacity sizing is a little complicated. In order to determine the battery capacity of a centralized PV-battery system, the concept of uniformity is applied[26]. Specifically, for a PV system without batteries, its uniformity (U-value) can be calculated according to the Eq. (1) as shown in Fig. 2a:

$$U = \frac{E_{gen}}{P_{max} \times \Delta t} \tag{1}$$

where $E_{gen}$ refers to the renewable generation; $P_{max}$ refers to the peak power of renewable generation in one day, and $\Delta t$ refers to 24 h in this research as shown in Fig. 2a.

After the battery is deployed, the energy output curve of the centralized PV-battery system will change as shown in Fig. 2b. Therefore, the changed U-value ($U'$) will be calculated according to the Eq. (2):

$$U' = \frac{E_{output}}{P'_{max} \times \Delta t} \tag{2}$$

where $E_{output}$ refers to the electricity output of the centralized PV-battery system; $P'_{max}$ refers to the peak power of the electricity output in one day, and $\Delta t$ refers to 24 h (one day) in this research as shown in Fig. 2a. Note that $P_{max}$ and $P'_{max}$ are different because the battery stores energy during periods of high power generation and discharges during periods of low power generation. $E_{gen}$ and $E_{output}$ are not the same. This is because there is a loss in battery charging and discharging.

Subsequently, the annual U-value of the system is obtained based on the average of the daily U-values. Based on the method shown in Eqs. (1–2), the relationship between U-value and battery capacity is obtained, as shown in Fig. 2c, where the relationship between U-value ($y$) and battery capacity ($x$) is fitted by the Eq. (3):

$$y = ax^3 + bx^2 + cx + d \tag{3}$$

The slope of the equation can also be found in Eq. (4):

$$y' = 3ax^2 + 2bx + c \tag{4}$$

The optimal battery capacity is identified at the largest slope in the mathematical relationship between U-value ($y$) and battery capacity ($x$). This is because that, the U-value reaches a peak as the battery capacity increases at the largest slope, which can achieve better results in energy storage, while the continuous increase in the battery capacity will reduce the U-value increase rate, and the battery can no longer perform its special functions.

Distributed prosumer-battery system: Considering the difference between the distributed PV-battery system and the centralized system, in

**Table 1 | Configuration of the reference case and other cases**

| Cases | Buildings | PV-battery system | Battery sizing method |
|---|---|---|---|
| Reference case | Hotel, Office, Residence | Not exist | N/A |
| Case 1 | Hotel, Office, Residence | Centralized | U-value |
| Case 2 | Hotel, Office, Residence | Distributed | U-value |
| Case 3 | Hotel, Office, Residence | Distributed | M-value |

addition to allowing the PV to supply stable electric energy, the distributed system also needs to consider the matching of the renewable generation and the electricity demand spatiotemporally. Therefore, we define a matching degree (M-value) calculated as shown in the Fig. 2b and Eqs. (5–7):

$$M_{\text{gen}} = \frac{E_{\text{surp}} + E_{\text{match}}}{E_{\text{surp}} + E_{\text{match}} + E_{\text{short}}} \tag{5}$$

$$M_{\text{dem}} = \frac{E_{\text{short}} + E_{\text{match}}}{E_{\text{surp}} + E_{\text{match}} + E_{\text{short}}} \tag{6}$$

$$M = (M_{\text{gen}} \times M_{\text{dem}})^{1/2} \tag{7}$$

where $E_{\text{surp}}$ refers to the surplus renewable energy of the distributed PV-battery system; $E_{\text{match}}$ refers to the self-consumption energy; $E_{\text{short}}$ refers to the demand shortage of the distributed prosumer-battery system.

When the battery is deployed, the matching degree (M-value) will be updated as shown in Eqs. (8–10):

$$M'_{\text{gen}} = \frac{E_{\text{surp}} + E_{\text{match}}}{E_{\text{surp}} + E_{\text{match}} + E_{\text{short}} - E_{\text{ch}}} \tag{8}$$

$$M'_{\text{dem}} = \frac{E_{\text{short}} + E_{\text{match}}}{E_{\text{surp}} + E_{\text{match}} + E_{\text{short}} - E_{\text{dis}}} \tag{9}$$

$$M' = (M'_{\text{gen}} \times M'_{\text{dem}})^{1/2} \tag{10}$$

where $E_{\text{ch}}$ and $E_{\text{dis}}$ refer to the energy charged and discharged to/from batteries. This method determines the required battery size based on matching the renewable generation curve and the building demand curve. Subsequently, the annual M-value of the system is obtained based on the average of the daily M-values. By using this method, the relationship between the M-value and battery capacities can be obtained as shown in Fig. 2d.

Note that the relationship between battery capacity and M-value shows the pattern of the S-curve in Fig. 2d, so they are fitted through the S-curve fitting equation as shown in Eq. (11):

$$y = \frac{a}{1 + be^{-cx}} \tag{11}$$

According to the NPV (Net present value) results in Fig. 2d, it is noted that the battery capacity represented by 1/2 of the maximum slope is very close to the battery capacity that maximizes the system NPV. Therefore, it can be considered that the battery capacity represented by 1/2 of the maximum slope in the mathematical relationship between M-value ($y$) and battery capacity ($x$) is the optimal battery capacity.

### Optimal sizing approach on an integrated building-photovoltaic-battery system (area-kWp-kWh)

Based on the above battery capacity sizing method, this study further proposes design guidelines and lifecycle carbon emission assessment in integrated building-PV-battery systems. An optimal design approach for integrated building-PV-battery energy systems in both centralized and distributed formats is proposed, including geographic locations, designed building areas, type and rated capacity of renewables, battery storage capacity in the form of m²-kWp-kWh relationship on a building-PV-battery system. Specifically, in a specific location with meteorological parameters, energy consumption of buildings can be obtained through smart metering or numerical simulation[30]. Based on the net-zero energy design principle[31], along with the local solar radiation resources and the dynamic photovoltaic efficiency, the required photovoltaic capacity can be calculated, and then the required battery capacity of the system can be further sized according to Fig. 2.

However, considering the huge gap between centralized systems and distributed systems, further discussion of the optimal system configuration and battery capacity sizing methods is necessary to derive the optimal area-kWp-kWh relationship. Therefore, the four cases shown in Table 1 are studied. The Reference case is not equipped with any renewable energy and energy storage systems. Therefore, there is no need for battery capacity sizing in the Reference case. Case 1 indicates that the case is a centralized PV-battery system, and battery sizing is performed through the commonly used U-value method (Fig. 2c). Case 2 represents the implementation of a distributed PV-battery system, but a new method for battery sizing has not yet been developed, so the traditional U-value method is still adopted (Fig. 2c). Case 3 adopts the proposed M-value method in the distributed PV-battery system (Fig. 2d).

In addition, in Cases 1, 2, and 3, the rated capacity of PV is designed to enable the local power generation in one year equal to its corresponding building energy demand. According to the above method, Fig. 3 shows the comparison between PV capacity, battery capacity, NPV and carbon intensity between different Cases in different buildings.

As shown in Figs. 3a–c, the Reference case is not equipped with PVs and batteries. In terms of PV configuration, the rated capacity of equipped PVs in Case 1 is slightly lower than that in Case 2 and Case 3. This is because that, Case 1 is a centralized system with designed tilted PVs at an optimal tilted angle. Case 2 and Case 3 are distributed systems with both tilted PVs (with an optimal tilted angle) and BIPVs (building-integrated PVs) with a vertical angle. The power generation of BIPVs is limited by their vertical angle and is much lower than that of the tilted PVs. Therefore, Case 2 and Case 3 need to be equipped with PVs in a larger capacity than that in Case 1.

In terms of battery capacity sizing, the capacity sizing of Case 2 is smaller than that of Case 1 and Case 3. The reason is below. Compared with Case 1, Case 2 is a distributed system, and the matching degree of renewable energy is higher than that of Case 1, resulting in a lower battery sizing capacity. Compared with Case 3, the U-value method used does not consider the mismatch between the demand side and the production side, resulting in an underestimation of the configured battery capacity. In addition, it is noted that the battery capacity required by the office is smaller than that of the residential and hotel buildings. This is because that, the energy demand of the office mainly occurs during the daytime, which is consistent with the PV power generation.

From an economic point of view, the gap between different Cases is small. However, it should be noted that overall, Case 3 can achieve a higher NPV than Case 1 and Case 2, indicating that the adopted capacity sizing method (M-value approach) in Case 3 shows higher economic feasibility than the traditional approach (U-value approach). In addition, the NPV of the residential building is much lower than that of the hotel building and

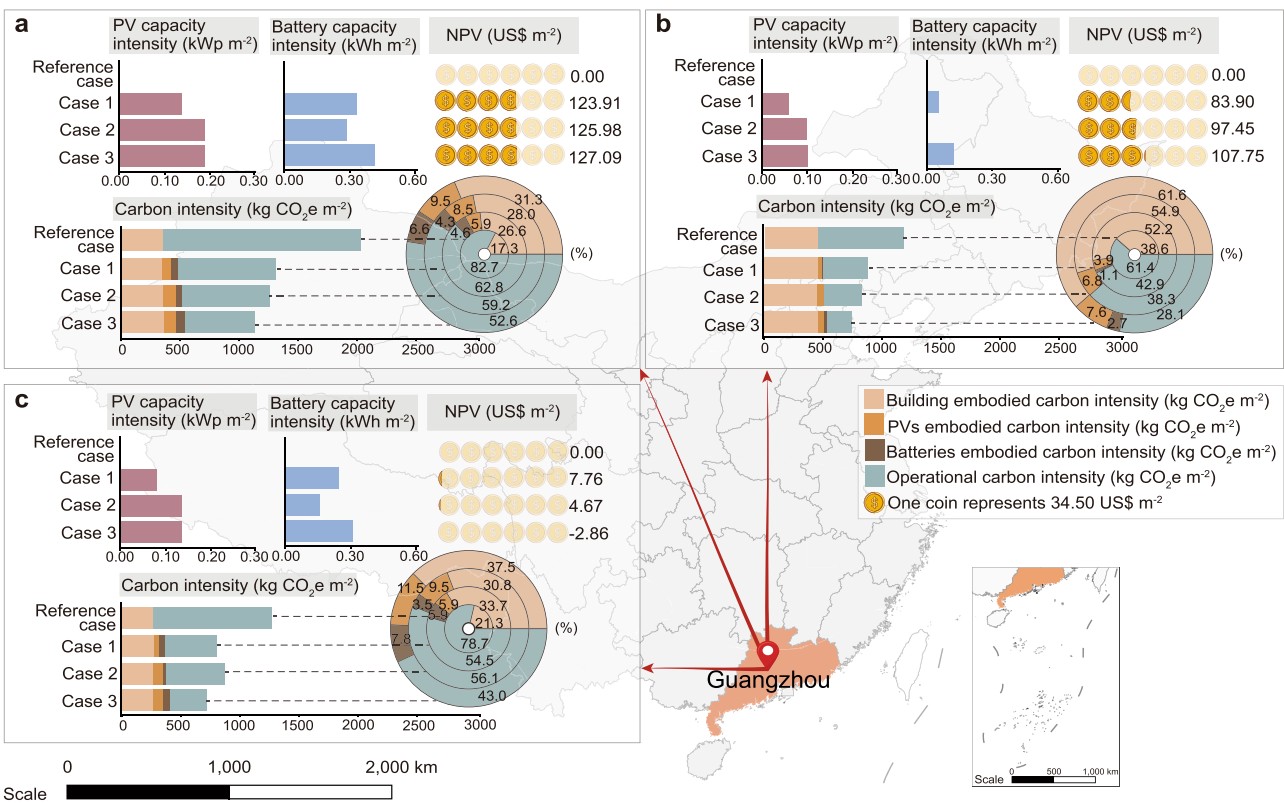

**Fig. 3 | Comparison of sized PV (photovoltaic) capacity, battery capacity, NPV (net present value) and carbon intensity for various types of buildings under different cases. a** Hotel buildings. **b** Office buildings. **c** Residential buildings.

office building. This is because that, the average electricity price used in the residential building (0.061 US$ kWh⁻¹) is lower than that of the hotel building and office building (0.083 US$ kWh⁻¹), resulting in lower import cost savings.

The lifecycle carbon intensity of the integrated building-PV-battery system needs to be quantified and compared to demonstrate the effectiveness of the system design approach on carbon mitigation. In terms of carbon emissions, the lifecycle carbon intensity analysis of the integrated building-PV-battery system shows that, the Reference Case shows the highest lifecycle carbon intensity, while PV panels and batteries in Cases 1, 2 and 3 can greatly decrease total carbon intensity. For the specific sub-items of carbon intensity, the embodied carbon of the buildings has basically remained unchanged in Cases 1, 2 and 3. The embodied carbon of PV and batteries increases slightly, but the operating carbon intensity is greatly reduced. For example, for the hotel building in Guangzhou, the embodied carbon of buildings in different cases is 350.71 kg $CO_2$e m⁻². The total embodied carbon of PV and batteries is 0, 139.03, 159.96, and 180.92 kg $CO_2$e m⁻² in Reference Case and Case 1, 2, and 3, respectively. The operational carbon intensity is 1670.79, 826.95, 740.23, and 590.21 kg $CO_2$e m⁻², respectively. Among them, Case 3 has the largest reduction in operating carbon intensity, indicating that the methods studied by Case 3 are the most suitable for achieving lower carbon emissions and moving toward carbon neutrality in the future.

In order to further analyse the total lifecycle carbon intensity of the integrated PV-battery-consumer system, the carbon emission structure in Case 3 is demonstrated in detail. As shown in Fig. 4a, even in the net-zero energy system, in addition to renewable energy sources, the distributed energy system in Case 3 also needs non-renewable energy and high-carbon emission energy sources for demand coverage (such as coal and natural gas). This is due to the spatiotemporal mismatch in renewables and energy demands. Correspondingly, carbon emissions as shown in Fig. 4b indicate that coal and natural gas account for about 99% of carbon emissions, and the

future energy structure needs to be further optimized to achieve carbon neutrality.

### Climate adaption and carbon footprint analysis in different climate zones in China

In Fig. 5, the calculated results are based on Guangzhou, which is located in a hot summer and warm winter area. In order to demonstrate the scalability of the proposed method in this study, this section is to analyse the feasibility of the approach in various areas with different climates. Therefore, in this section, based on the M-value battery sizing method used in Case 3, the area-kWp-kWh relationship, NPV and lifecycle carbon intensity are analysed for five climate zones in China.

Relevant results for cities in different climate zones across China are shown in Fig. 5, including Beijing (Fig. 5a), Shenyang (Fig. 5b), Kunming (Fig. 5c), Shanghai (Fig. 5d), Guangzhou (Fig. 5e) and Shenzhen (Fig. 5f). Results show that the PV capacity and battery capacity required by the hotel are the largest, followed by the residential building and office building, respectively. This is due to the high energy demand of the hotel building. This also results in the hotel building's carbon intensity being higher than that of other buildings. In addition, due to the high commercial electricity prices, the hotel building and office building can obtain considerable import cost savings and achieve high NPVs, while the residential building is restrained by its low electricity prices and low import cost savings, so its NPVs are low. The comparison between different climate zones indicates that the lifecycle carbon intensities of the well-designed photovoltaic-battery-consumer energy systems in hot summer and warm winter zones, mild climate zones, and hot summer and cold winter zones are relatively low, while lifecycle carbon intensities in cold zones and severe cold zones are high. This is because that, electricity demand in cold and severe cold zones is mainly concentrated in winter, while PV generation is mainly concentrated in summer, resulting in a seasonal mismatch between demand and renewable electricity supply, which cannot be covered by batteries alone.

**Fig. 4 | Energy flow and carbon flow analysis in the distributed PV-battery-consumer system (Case 3).** **a** Energy flow. **b** Carbon flow.

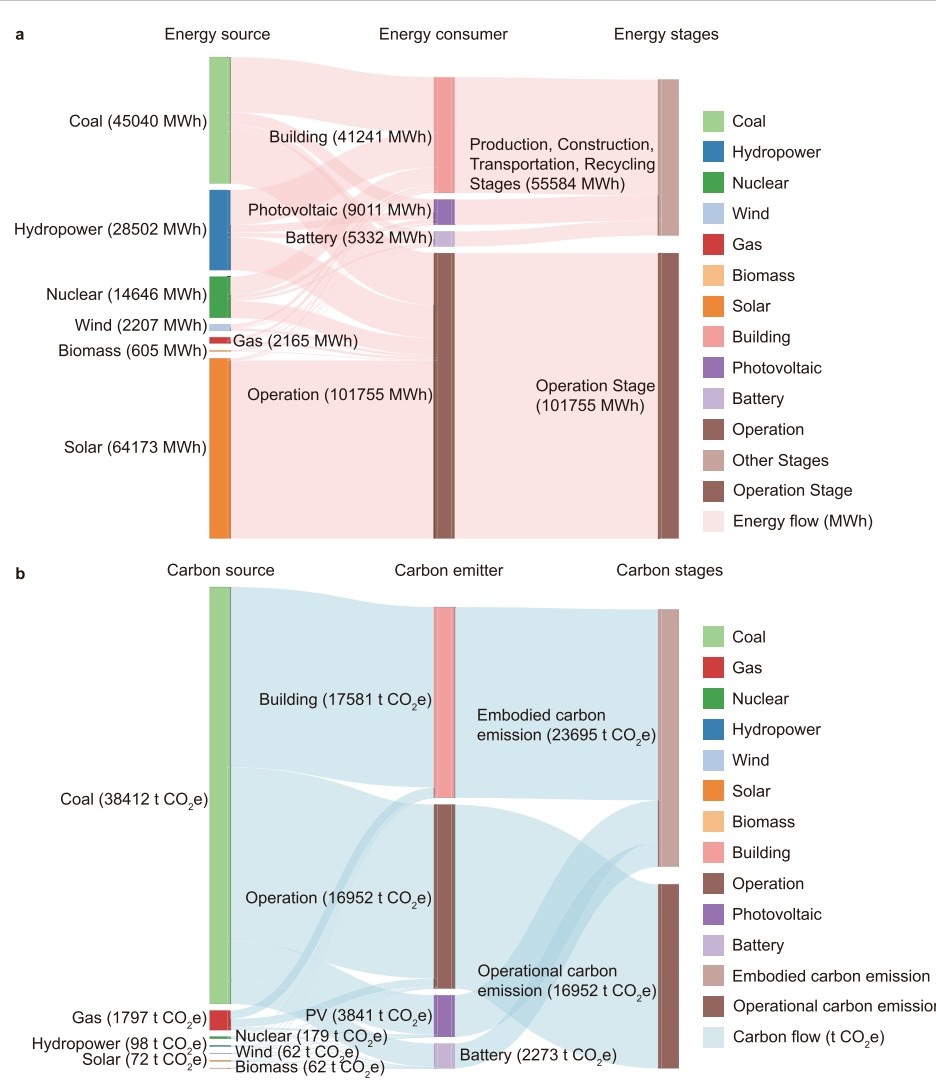

Through the above analysis, it can be proved that the proposed design guidelines and lifecycle carbon intensity analysis method can be widely applied in net-zero energy paradigms of building-photovoltaic-battery distributed energy systems, and are suitable for cities and regions under different climate conditions. This can provide recommendations for sustainability policy formulation and the achievement of dual carbon targets.

**Comparison on different battery capacity sizing methods in distributed systems**

To further illustrate the rationality of the M-value method for battery capacity sizing in distributed systems, the results are compared with previous studies[32], which uses multi-objective optimization to determine the optimal battery capacity for hotel, office, and residential buildings based on maximizing economic and environmental benefits. The optimal battery capacities are found to be 895.6 kWh, 900 kWh and 895.6 kWh (which means a battery capacity intensity of 0.0746, 0.0750 and 0.0746 kWh m$^{-2}$ based on an area of 12,000 m$^2$), respectively[32]. When these battery capacity intensities are applied to Guangzhou, the results showed an overall lower net present value (NPV) and higher carbon intensity compared to the method used in this study as shown in Fig. 6a–c. Specifically, the NPV for the hotel building is 93.70 US$ m$^{-2}$, which is much lower than 127.09 US$ m$^{-2}$ for the results in this study. Furthermore, the carbon intensities for the research method from Ref. [32] are 1526.94, 756.37, and 987.75 kg CO$_2$e m$^{-2}$, respectively, while they are higher than 1121.84, 736.20 and 720.59 kg CO$_2$e m$^{-2}$ in this study. Therefore, it can be noticed that the method proposed in this

study has more advantages than traditional approaches based on maximizing economic and environmental benefits in NPV and carbon emissions.

## Conclusions

In this work, considering the uncertainty and complexity of optimal design, an optimal design approach with cross-climate adaption for integrated photovoltaic-battery-consumer energy systems is proposed to avoid poor techno-economic-environmental performance. Lifecycle carbon intensity is quantified in the integrated photovoltaic-battery-consumer energy system, considering the embodied carbon emissions in both renewable systems and battery materials, operational carbon emissions, and dynamic performance degradation and replacement times. Design guidelines based on the building area-renewable capacity-battery capacity with m$^2$-kWp-kWh relationship and net-zero energy principle are proposed, including the configuration of PV and battery capacity for different building types and economic and environmental impact analysis under different configuration scenarios. Superiorities of the proposed matching degree (M-value) approach are verified, through the comparison with the traditional uniformity (U-value) approach, in terms of PV capacity, battery capacity, net present value and lifecycle carbon intensity. Furthermore, the proposed method is scalable, extended and applied to five climate zones in China with different solar resources and climate conditions to guide the economic-environmental feasibility of energy, social, and governance investment behaviours. Research results can be applied in both centralized and distributed

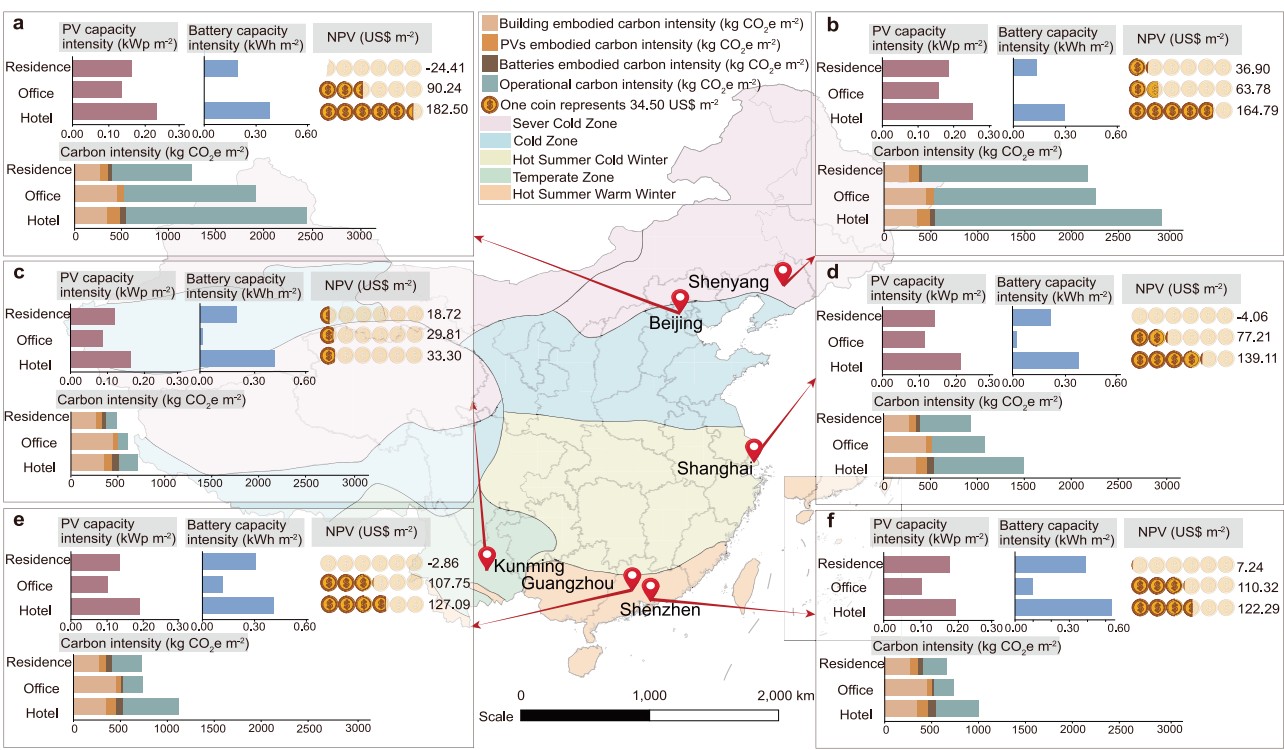

**Fig. 5 | PV (photovoltaic) capacity, battery capacity, NPV (net present value) and carbon intensity for different cities. a** Beijing. **b** Shenyang. **c** Kunming. **d** Shanghai. **e** Guangzhou. **f** Shenzhen.

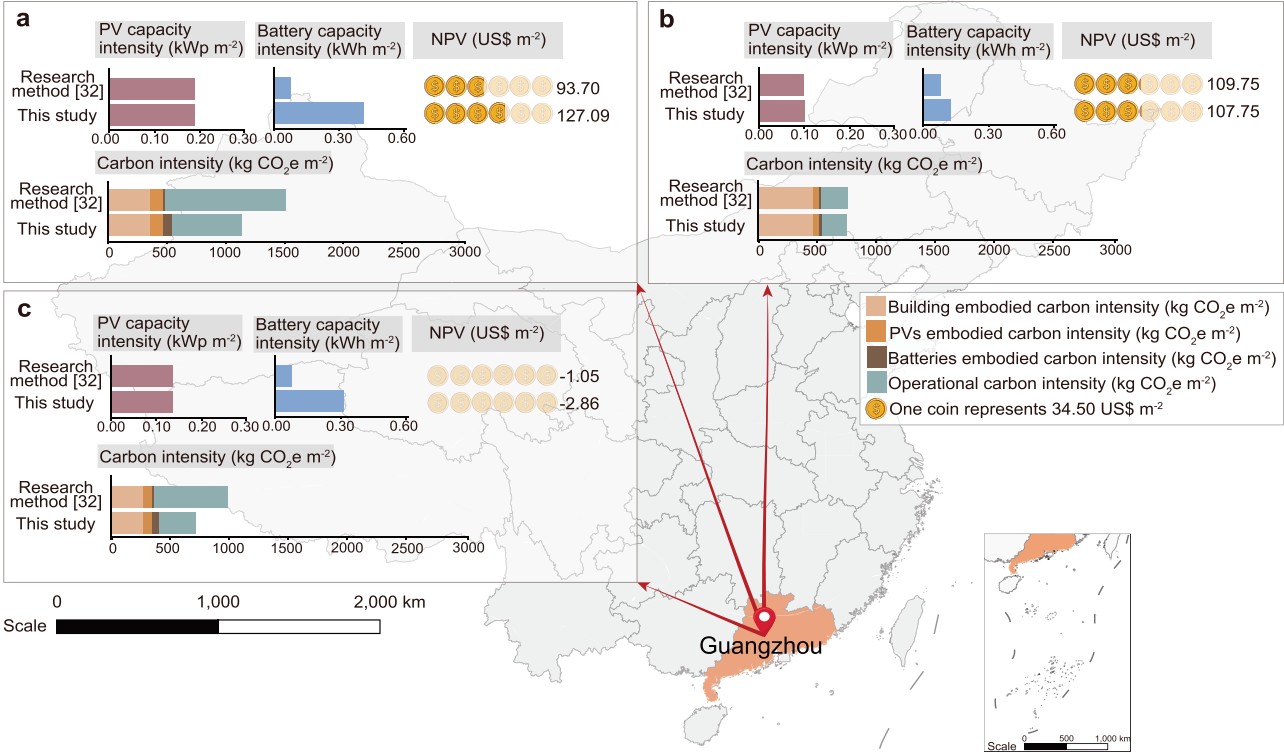

**Fig. 6 | Comparison of different battery sizing approaches in each building in terms of PV (photovoltaic) capacity, battery capacity, NPV (net present value) and carbon intensity. a** Hotel buildings. **b** Office buildings. **c** Residential buildings.

photovoltaic-battery-consumer energy systems for Sustainable Development Goals and carbon neutrality transition. The specific conclusions of this work are as follows:

(1) Unlike the battery storage capacity design on centralized PV-battery-consumer systems that focus primarily on stable electricity supply, distributed prosumer-battery energy system design needs to consider both

dynamic renewables, electricity demand, and their matching spatio-temporally. The traditional uniformity (U-value) approach will lead to battery undersizing or oversizing, and resource waste, together with poor economic and environmental performance, while the proposed matching degree (M-value) approach shows robustness and superiorities over the traditional uniformity (U-value) approach. In the studied case, for integrated building-PV-battery system design of a hotel building in Guangzhou (subtropical climate), compared to the traditional uniformity (U-value) approach, the proposed matching degree (M-value) approach can increase the net present value from 123.91 to 127.09 US$ m$^{-2}$ (by 2.57%) and reduce lifecycle carbon intensity from 1316.69 to 1121.84 kg $CO_2$e m$^{-2}$ (by 14.80%), respectively.

(2) Design guidelines for photovoltaic-battery-consumer energy systems are proposed following the net-zero energy principle, to enable the techno-economic-environmental performance optimality. Based on the proposed matching degree (M-value) approach, the m$^2$-kWp-kWh relationship of the building-PV-battery system is obtained in different climate regions in China to guide the integrated energy system design. For example, in a distributed system, the relationship of a hotel building in Beijing, Guangzhou, Kunming, Shanghai, Shenyang, Shenzhen is 1 m$^2$-0.2323 kWp-0.3801 kWh, 1 m$^2$-0.1906 kWp-0.4111 kWh, 1 m$^2$-0.1623 kWp-0.4106 kWh, 1 m$^2$-0.2116 kWp-0.3758 kWh, 1 m$^2$-0.2553 kWp-0.2948 kWh, 1 m$^2$-0.1923 kWp-0.5309 kWh, respectively. Under the guidance of this design guideline, the renewable penetration rate, economic and environmental performance can be increased.

(3) The analysis of different climate zones demonstrates the climate adaptability of the energy system design guidelines. The temperature, humidity, building design, and solar resources vary greatly among the five climate zones in China, resulting in different area-kWp-kWh relationships. It is noteworthy that the PV capacity design and carbon intensity are lower in hot summer and warm winter zones, hot summer and cold winter zones, and mild climate zones, while they are higher in cold and severe cold zones.

The conclusions of this paper propose effective design guidelines for distributed energy systems to achieve carbon neutrality, providing useful suggestions for decentralized, renewable, and sustainable development, clean grid transformation, and circular economy. However, this paper still has limitations: (1) Other types of renewable energy are not considered in this paper, so how to match suitable energy storage systems in the case of mixed renewable energy (such as wind energy, tidal energy, and geothermal energy) is still a challenge. (2) The interaction between building energy systems and electric vehicles is not considered in this paper, and the participation of electric vehicles in the design guidelines of distributed energy systems is also worth studying. Therefore, future work will focus on complex energy systems with multiple renewable energy sources and vehicle-to-everything interaction and provide corresponding energy system design guidelines.

## Methods
### Circular economic framework
In order to explore and calculate the lifecycle cost and carbon emissions of the building-PV-battery energy storage integrated system, this study established a calculation model by TRNSYS 18 based on the circular economy framework[33].

The circular economy framework includes building circular economy, photovoltaic circular economy and battery circular economy.

The building circular economy is divided into four phases: raw material manufacturing, transportation and construction, operation and maintenance, demolition and reuse. Among them, the carbon emission generated in the operation and maintenance phase is the operation carbon emission, while the carbon emissions in the raw material, manufacturing, transportation and construction, demolition and reuse phases are called embodied carbon emissions. Based on a cradle-to-grave lifecycle analysis, the lifecycle embodied carbon emissions of hotels, office buildings and residences are 701.42, 906.93 and 541.14 kg $CO_2$e m$^{-2}$, respectively[34].

The PV circular economy includes four phases, raw material and processing phase, production and installation phase, operation phase, demolition and recycling phase. The embodied carbon emissions of the three stages including raw materials and processing, production and installation, demolition and recycling are 560 kg $CO_2$e kWp$^{-1}$[35].

With respect to the battery circular economy[36], it is divided into four phases: materials mining, manufacturing, operation and reuse, recycling. In the material mining phase, the lifecycle embodied carbon emission of the materials used to produce LiFePO$_4$ batteries are 51 kg $CO_2$e kWh$^{-1}$[37]. During the manufacturing phase, batteries are manufactured and then used in buildings as the energy storage system with a lifecycle embodied carbon emission of 34 kg $CO_2$e kWh$^{-1}$[37].

In addition, the weather data used for system simulation are shown in Supplementary Fig. 1, and the PV parameters are shown in Supplementary Table 1.

### Lithium-ion battery degradation model
The battery model used in this paper is extended from a previously developed battery model[38]. Supplementary Fig. 2 shows the degradation curve of the battery model, which is characterized by the different degradation speeds of the battery under different depths of discharge (DoD). According to this characteristic, a system based on the number of cycles and dynamic DoD is developed. The dynamic DoD in Li-ion batteries (LFP batteries) can be calculated by Eq. (12):

$$DoD = FSOC_{peak,cycle,n} - FSOC_{valley,cycle,n} \qquad (12)$$

where $FSOC_{peak,cycle,n}$ and $FSOC_{valley,cycle,n}$ refer to the highest and the lowest point of the fractional state of charge (FSOC) at the $n^{th}$ cycle.

In addition, according to Supplementary Fig. 2, relative capacity (RC) is used to illustrate the degree of degradation of lithium-ion batteries. The Li-ion battery degradation curve can be polynomial fitted with exponential power at 3[39], which is expressed in Eq. (13).

$$RC_{DoD} = k_1 CycleNum^3 + k_2 CycleNum^2 + k_3 CycleNum + k_4 \qquad (13)$$

where $k_1$ to $k_4$ are polynomial coefficients, and CycleNum is the number of cycles. The values from $k_1$ to $k_4$ under different DoDs are listed in Supplementary Table 2. Therefore, the dynamic battery degradation is calculated by Eqs. (14–17):

$$\Delta CycleNum_{(1or2),t+\tau} = \frac{E_{t \to t+\tau}}{Cap_t \times 2 \times DoD_{1or2}} = \frac{|FSOC_{t+\tau} - FSOC_t| \times Cap_t}{Cap_t \times 2 \times DoD_{1or2}} = \frac{|FSOC_{t+\tau} - FSOC_t|}{2 \times DoD_{1or2}} \qquad (14)$$

$$CycleNum_{(1or2),t+\tau} = CycleNum_{(1or2),t} + \Delta CycleNum_{(1or2),t+\tau} \qquad (15)$$

$$RC_{DoD1 \, or \, DoD2,t+\tau} = k_1 CycleNum_{(1or2),t+\tau}{}^3 + k_2 CycleNum_{(1or2),t+\tau}{}^2 + k_3 CycleNum_{(1or2),t+\tau} + k_4 \qquad (16)$$

$$RC_{DoD,t+\tau} = \frac{DoD_{t+\tau} - DoD_2}{DoD_1 - DoD_2} \times RC_{DoD1,t+\tau} + \frac{DoD_{t+\tau} - DoD_1}{DoD_2 - DoD_1} \times RC_{DoD2,t+\tau} \qquad (17)$$

where the subscripts 1 and 2 refer to two adjacent curves close to the time-variant DoD; $\Delta CycleNum_{t+\tau}$ refers to the increase in number of cycles from time $t$ to time $t + \tau$; $E_{t \to t+\tau}$ refers to the total energy charged and discharged from time $t$ to time $t + \tau$; $Cap_t$ refers to the time-variant capacity of the battery at time $t$; $DoD_1$ and $DoD_2$ refer to the DoD of two curves adjacent to the actual continuous DoD; $FSOC_t$ and $FSOC_{t+\tau}$ refer to the FSOC at time $t$ and time $t + \tau$; $CycleNum_t$ and $CycleNum_{t+\tau}$ refer to the number of cycles of batteries at time $t$ and time $t + \tau$; $RC_{DoD}$, $RC_{DoD1}$ and $RC_{DoD2}$ refer to the dynamic relative capacity under actual DoD, $DoD_1$ and $DoD_2$; $DoD_{t+\tau}$

refers to the actual DoD at time $t + \tau$. Simulation time step $\tau$ is 1 hour in this study.

Equation (14) is used to calculate the increased number of cycles at each time step. Equation (15) is used to calculate the number of battery cycles at the current time step, Eq. (16) is used to calculate the relative capacity corresponding to the current number of battery cycles under curve 1 and curve 2, and Eq. (17) calculates the relative capacity under the real DoD by interpolation.

In addition, the battery's maximum charging and discharging power is limited to 0.5 C, indicating that the battery can be charged to its full capacity within a maximum of 2 h.

## Assessment criteria

The net present value (NPV) is calculated based on Eq. (18):

$$\text{NPV} = \Delta C_{\text{imp,save}} - \Delta C_{\text{recyc}} - \Delta C_{\text{remanu}} - \Delta C_{\text{O\&M}} - \Delta IC_{\text{PV}} - \Delta IC_{\text{bat}} \tag{18}$$

The reference case and the studied case are as shown in Supplementary Fig. 3. Note that $\Delta$ in Eq. (18) represents the difference between the studied case and the reference case. The $C_{\text{imp,save}}$ refers to the cost saving from importing grid electricity (Supplementary Tables 3–14), and the calculation needs to consider an annual escalation rate of 1.4%[40]. Note that the renewable energy system in this study does not sell electricity to the grid. This is due to the lack of unified coordination between China's photovoltaic power generation and power grid construction[41], as well as the slow introduction of local policies[42]. Therefore, a conservative assumption is made in the study. $C_{\text{recyc}}$ refers to the recycling cost of EoL (End-of-Life) batteries (with RC at 60%) (57 \$ kWh$^{-1}$)[43]. $C_{\text{remanu}}$ refers to the cost of remanufacturing new batteries from recycled batteries (40 \$ kWh$^{-1}$)[44]. $C_{\text{O\&M}}$ refers to the operating and maintenance cost of subcomponents (like batteries, BIPVs and Solar PVs). The annual operation and maintenance (O&M) cost for batteries is 0.5% of the initial investment cost. For BIPVs and Solar PVs, the annual O&M cost is 5% of the initial cost[45–49]. $IC_{\text{PV}}$ is the initial investment of BIPVs and Solar PVs, which is 240 US\$ kWp$^{-1}$ (kWp means kilowatt peak power). $IC_{\text{bat}}$ refers to the initial investment cost of batteries (139 \$ kWh$^{-1}$)[50]. The exchange rate between Chinese Yuan and US Dollar is 1 CN¥ = 0.1429 US\$. The annual interest rate is 2%[40]. More specific calculation Equations are shown in Supplementary Note 1.

This paper unifies the calculation methods of carbon emissions and carbon intensity based on the concept of building-PV-battery storage integration. The lifecycle carbon emissions of buildings, PV panels and batteries can be calculated in four stages: raw materials, production (including transportation and construction), operation, and recycling (including dismantling and reuse). Among them, the carbon emissions generated in the raw material, production and recycling stages are embodied carbon emissions. Therefore, the embodied carbon emissions (ECE) of buildings, photovoltaic panels and batteries can be calculated by Eqs. (19–21):

$$\text{ECE}_{\text{bld}} = \text{CE}_{\text{raw,bld}} + \text{CE}_{\text{con,bld}} + \text{CE}_{\text{trans,bld}} + \text{CE}_{\text{dism,bld}} \tag{19}$$

$$\text{ECE}_{\text{PV}} = \text{CE}_{\text{raw,PV}} + \text{CE}_{\text{prod,PV}} + \text{CE}_{\text{recyc,PV}} \tag{20}$$

$$\text{ECE}_{\text{bat}} = \text{CE}_{\text{raw,bat}} + \text{CE}_{\text{prod,bat}} + \text{CE}_{\text{recyc,bat}} \tag{21}$$

where the subscripts bld, PV, and *bat* represent buildings, PV panels, and batteries, respectively. The subscripts raw, con, prod, trans, dism, and recyc respectively represent the raw materials, construction, production, transportation, dismantling and recycling stages, respectively.

For buildings[34],

Carbon emission (CE) of the Hotel building: 658.98 kg $CO_2$e m$^{-2}$ for raw materials stage, 25.80 kg $CO_2$e m$^{-2}$ for construction stage, 14.11 kg $CO_2$e

m$^{-2}$ for transportation stage, 2.52 kg $CO_2$e m$^{-2}$ for dismantling stage. Total in 701.42 kg $CO_2$e m$^{-2}$.

Carbon emission of the Office building: 862.78 kg $CO_2$e m$^{-2}$ for the raw materials stage, 25.80 kg $CO_2$e m$^{-2}$ for the construction stage, 38.70 kg $CO_2$e m$^{-2}$ for the transportation stage, 2.52 kg $CO_2$e m$^{-2}$ for dismantling stage. Total in 906.93 kg $CO_2$e m$^{-2}$.

Carbon emission of the Residential building: 501.63 kg $CO_2$e m$^{-2}$ for the raw materials stage, 25.80 kg $CO_2$e m$^{-2}$ for construction stage, 11.18 kg $CO_2$e m$^{-2}$ for transportation stage, 2.52 kg $CO_2$e m$^{-2}$ for dismantling stage. Total in 541.14 kg $CO_2$e m$^{-2}$.

Note that the hotel building has 30 floors, and each floor is 400 m$^2$. The office building has 30 floors, and each floor is 625 m$^2$. The residential building has 30 floors, and each floor is 600 m$^2$. For PVs, $\text{ECE}_{\text{pv}}$ is 560 kg $CO_2$e kWp$^{-1}$[35,51].

For batteries, $\text{CE}_{\text{raw,bat}}$ is equal to 51 kg $CO_2$e kWh$^{-1}$ and $\text{CE}_{\text{prod,bat}}$ is equal to 34 kg $CO_2$e kWh$^{-1}$[37]. The $\text{CE}_{\text{recyc,bat}}$ is equal to 69.8 kg $CO_2$e kWh$^{-1}$[37].

In the operation stage, the carbon emission is calculated based on Eq. (22):

$$\text{CE}_{\text{ope}} = \int_0^{t_{\text{end}}} P_{\text{grid}}(t) \cdot \text{CEF}_{\text{eg}} dt \tag{22}$$

where $t_{\text{end}}$ refers to the end time of the simulation (25 years in this study), $P_{\text{grid}}(t)$ refers to the power of grid imported electricity, and $\text{CEF}_{\text{eg}}$ refers to the carbon emission factor of the local electricity grid.

## Data availability

The authors declare that the data used as model inputs supporting the findings of this study are available within the paper and its Supplementary Information files. In addition, all data and calculation tables have been uploaded to https://zenodo.org/records/14037792.

## Code availability

The authors declare that they used TRNSYS 18 software for modelling. The TRNSYS codes are available from the corresponding author upon request.

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

## Acknowledgements

This work was supported by National Natural Science Foundation of China (NSFC) (52408137), Guangdong Provincial Natural Science Foundation General Project (2414050003253, 2024A1515011609), Guangdong Province Joint Fund for Basic and Applied Basic Research (2022A1515110364, 2023A04J1035), Municipal University (Institute) Enterprise Joint Funding Project (2023A03J0104), Yangcheng Scholars Leading Talent Training Project (2024312133), HKUST(GZ)-enterprise cooperation projects (R00017-2001, R00072-2001, R00079-2001). This research is also supported by The Hong Kong University of Science and Technology (Guangzhou) startup grant (G0101000059). This work was supported in part by the Project of Hetao Shenzhen-Hong Kong Science and Technology Innovation Cooperation Zone (HZQB-KCZYB-2020083).

## Author contributions

Aoye Song: Conceptualization, Methodology, Software, Investigation, Writing – Original Draft. Yuekuan Zhou (Corresponding Author): Conceptualization, Methodology, Writing – Review & Editing, Funding Acquisition, Supervision.

## Competing interests

The authors declare no competing interests.
