## [Peer Review file · Communications Engineering]

An Integrative lifecycle design approach for renewable-battery-consumer energy systems

Corresponding Author: Dr Yuekuan ZHOU

Version 0:

Reviewer comments:

Reviewer #1

(Remarks to the Author)

The paper provides a strong rationale for the presented research and effectively outlines the gaps in the literature regarding battery capacity sizing. The analysis is sound, and the presentation of the work is commendable. However, I have the following observations:

(1) I found some confusion arising from the statement "Until now, there is no scientific methodology in sizing battery capacity...". This assertion overlooks several published works in this research domain. For instance:

S. A. P. Kani, P. Wild and T. K. Saha, "Improving Predictability of Renewable Generation Through Optimal Battery Sizing," in IEEE Transactions on Sustainable Energy, vol. 11, no. 1, pp. 37-47, Jan. 2020, doi: 10.1109/TSTE.2018.2883424.

A. Z. Gabr, A. A. Helal and N. H. Abbasy, "Multiobjective Optimization of Photo Voltaic Battery System Sizing for Grid-Connected Residential Prosumers Under Time-of-Use Tariff Structures," in IEEE Access, vol. 9, pp. 74977-74988, 2021, doi: 10.1109/ACCESS.2021.3081395.

M. Aghamohamadi, A. Mahmoudi and M. H. Haque, "Two-Stage Robust Sizing and Operation Co-Optimization for Residential PV-Battery Systems Considering the Uncertainty of PV Generation and Load," in IEEE Transactions on Industrial Informatics, vol. 17, no. 2, pp. 1005-1017, Feb. 2021, doi: 10.1109/TII.2020.2990682.

A. Shankar, K. Vijayakumar, B. C. Babu and R. Kaur, "Energy Trilemma Index-Based Multiobjective Optimal Sizing of PV-Battery System for a Building in Tropical Savanna Climate," in IEEE Systems Journal, vol. 16, no. 4, pp. 5630-5638, Dec. 2022, doi: 10.1109/JSYST.2022.3167166.

L. Bhamidi and S. Sivasubramani, "Optimal Sizing of Smart Home Renewable Energy Resources and Battery Under Prosumer-Based Energy Management," in IEEE Systems Journal, vol. 15, no. 1, pp. 105-113, March 2021, doi: 10.1109/JSYST.2020.2967351.

These are merely a selection of examples, and there exist numerous other papers addressing battery capacity sizing while considering renewable energy intermittency. Hence, the authors must elucidate the novelty of their work more clearly.

(2) It would be beneficial to elaborate on the U-value and M-value approaches utilized in this study for capacity sizing. While the U-value traditionally pertains to assessing a product's insulation properties, its adaptation for this research warrants further justification.

(3) I suggest including a graphical representation to demonstrate how the proposed Circular Economy Framework was applied in this study for clarity and better understanding.

Reviewer #2

(Remarks to the Author)

The paper presents a design approach for integrated PV-battery-consumer energy systems aimed at battery capacity sizing. Lifecycle carbon intensity for the integrated system is quantified, and a design guideline based on area-renewable capacity-battery capacity is provided. While the paper has some merits, its contributions are marginal. The structure of the paper is fragmented, which makes it difficult to follow the story. More comments as follows:

- The research motivations are not clear. The authors discuss the battery sizing capacity as the main motivation. However, it is not clear what are the research questions that haven't been addressed in the literature.
- What does the statement "there is currently a lack of research on the renewable energy penetration rate" mean, and how is it related to this research?
- The difference between centralized PV-battery-consumer and distributed systems is not well explained. If the authors are referring to residential PV-battery systems, how can it be considered a centralized system?
- The contributions of the work are marginal. The paper is more focused on doing an analysis of battery sizing and impact of different metrics rather than proposing a new methodology which improves the existing methods in the literature.
- Why are the results and discussion presented right after the introduction before explaining the methodology?
- Equations should be explained in the main body not in the figures.
- The paper needs to be restructured. It is difficult to follow different sections.

Reviewer #3

(Remarks to the Author)

Review report

This study focuses on carbon neutrality, instead of technical and economic aspects, of the PV-BESS-Prosumer set. It aims to develop a methodology for integrated system capacity sizing with lifecycle carbon intensity quantification and net present value for techno-economic environmental performance optimality. The study proposes low-carbon design guidelines for integrated PV-battery-consumer energy systems with climate adaptability, to assist decision making among planners, designers, managers and policy-makers.

The subject is timely, the proposed idea is practically interesting and the methodology sounds. However, I have still concerns about clarity of the paper; some of them are:

1- Most of the existing battery sizing optimization techniques consider economic and technical objectives to achieve the best using minimum battery size. They mostly consider a constraint on battery depth of charge and also on overcharging; thus, battery degradation is also considered. My understanding is that using those methods, with minimum BESS size and low degradation, the best option in terms of carbon neutrality in production and recycling has already been achieved (although it may not be directly claimed). Therefore, I have a question about the real contribution of the paper. Authors can clarify their contribution by comparing the results of their method with one recent literature with technical or economic objectives.

2- The proposed methodology assumes that all renewable generation must be either consumed or stored in BESS, at all times. Although a reasonable assumption for grid planning, this methodology may cause too conservative results considering that the power grid can tolerate the specific amount of extra generation based on available standards, e.g. IEC60038. This assumption can reduce the size of required BESS, both in centralised and decentralised cases; thus contributing to carbon neutrality.

Best regards,

Ali Moradi Amani

School of Engineering

RMIT University, Melbourne, Australia

Version 1:

Reviewer comments:

Reviewer #1

(Remarks to the Author)

The authors have addressed all my comments and made the necessary changes in the manuscript. Thank you for that. I do not have any further comment.

Reviewer #2

(Remarks to the Author)

Thanks for addressing my comments. The paper has been improved significantly and can be accepted.

Reviewer #3

(Remarks to the Author)

Authors have clarified my comments in the articles. I have no further comments.

Detailed reply to the reviewers

1st Revision (Apr 2024)

Specific reply to the comments one by one:

Reviewer #1 (Remarks to the Author): The paper provides a strong rationale for the presented research and effectively outlines the gaps in the literature regarding battery capacity sizing. The analysis is sound, and the presentation of the work is commendable. However, I have the following observations:

(1) I found some confusion arising from the statement "Until now, there is no scientific methodology in sizing battery capacity...". This assertion overlooks several published works in this research domain. For instance:

S. A. P. Kani, P. Wild and T. K. Saha, "Improving Predictability of Renewable Generation Through Optimal Battery Sizing," in *IEEE Transactions on Sustainable Energy*, vol. 11, no. 1, pp. 37-47, Jan. 2020, doi: 10.1109/TSTE.2018.2883424.

A. Z. Gabr, A. A. Helal and N. H. Abbasy, "Multiobjective Optimization of Photo Voltaic Battery System Sizing for Grid-Connected Residential Prosumers Under Time-of-Use Tariff Structures," in *IEEE Access*, vol. 9, pp. 74977-74988, 2021, doi: 10.1109/ACCESS.2021.3081395.

M. Aghamohamadi, A. Mahmoudi and M. H. Haque, "Two-Stage Robust Sizing and Operation Co-Optimization for Residential PV–Battery Systems Considering the Uncertainty of PV Generation and Load," in *IEEE Transactions on Industrial Informatics*, vol. 17, no. 2, pp. 1005-1017, Feb. 2021, doi: 10.1109/TII.2020.2990682.

A. Shankar, K. Vijayakumar, B. C. Babu and R. Kaur, "Energy Trilemma Index-Based Multiobjective Optimal Sizing of PV-Battery System for a Building in Tropical Savanna Climate," in *IEEE Systems Journal*, vol. 16, no. 4, pp. 5630-5638, Dec. 2022, doi: 10.1109/JSYST.2022.3167166.

L. Bhamidi and S. Sivasubramani, "Optimal Sizing of Smart Home Renewable Energy Resources and Battery Under Prosumer-Based Energy Management," in *IEEE Systems Journal*, vol. 15, no. 1, pp. 105-113, March 2021, doi: 10.1109/JSYST.2020.2967351.

These are merely a selection of examples, and there exist numerous other papers addressing battery capacity sizing while considering renewable energy intermittency. Hence, the authors must elucidate the novelty of their work more clearly.

Reply: Thank you very much for your very professional comment. The previous statement "Until now, there is no scientific methodology in sizing battery capacity..." in the article is indeed slightly exaggerated. After carefully checking the literature, we notice the following facts:

1. Currently, most battery sizing methods are based on the PV generation side. For example, the methods mentioned in Ref. [12, 13, 26] only consider the PV generation side, which is not conducive to sizing the battery capacity in distributed energy systems. This is because batteries in distributed energy systems

need to balance on-site renewable energy generation and energy demand simultaneously. If the method lacks consideration for energy demand, it will result in deviation in battery capacity sizing, leading to resource wastage, reduced revenue, and environmental pollution. Therefore, a method that considers both generation and consumption sides (such as buildings and EVs) is essential.

2. Some methods that consider both generation and consumption sides introduce economic and environmental factors, which are not very reliable due to the uncertainty in prices and carbon emissions. For example, Ref. [10] and [A] only consider economic and environmental factors. Only considering economic and environmental indicators may result in unreasonable battery capacity sizing, as net present value (NPV) and carbon emissions are highly dependent on various independent variables (such as renewable energy costs, battery prices, electricity prices, carbon emission factors, and embodied carbon emissions) and various intermediate variables (such as renewable penetration rates, demand shortages, battery charge and discharge power, and battery degradation rates). Therefore, the current battery sizing will be underestimated or overestimated in the future due to price fluctuations and improvements in technology levels. Moreover, battery capacity sizing based on economic and environmental indicators cannot reveal the basic physical mechanisms of dynamic interconnection and power interaction within the integrated PV-battery-consumer energy system. Therefore, a battery capacity sizing method that can satisfy the dynamic balance between the generation and consumption sides is necessary.

Based on these facts, we changed it to the following statement: *‘Up until now, there is a lack of appropriate methods for sizing battery capacity in distributed energy systems, and inappropriate methods can result in the under-sizing or over-sizing of battery capacity, material waste and economic-environmental infeasibility.’* (Section **Introduction**)

In addition, we greatly appreciate your provision of five papers on battery sizing for our reference. Regarding your concerns about the novelty of our paper, we have analysed the differences between our study and papers you listed, as detailed below:

Ref [25] (The first paper provided) proposes an optimal battery sizing method based on the concept of time series predictability. The main difference between this paper and ours is that Ref [25] mainly focuses on optimizing PV-battery systems, while our paper considers the impact of the demand-side.

Ref [20, 21, 22] (The second, third and fourth papers provided) focuses on renewable energy system sizing for residential PV-battery systems. The difference between their works and ours is that their optimization process does not consider the dynamic degradation of PV generation and battery capacity, which can cause significant deviation in battery sizing.

Ref [11] (The fifth paper provided) optimizes renewable energy and battery sizing for smart home renewable energy resources and battery systems. The difference between this paper and our study is that the two optimization objectives proposed in this paper include the minimum total lifecycle cost of batteries, the minimum daily energy cost of consumers, and the maximum total comfort index of consumers. The article does not discuss a method that considers technical indicators (such as renewable penetration, demand shortage, battery charge and discharge power, and battery degradation rates) and

lacks in-depth research on the dynamic interconnection and power interaction within the PV-battery-consumer system.

Based on the above analysis, we cited 5 papers you provided (Citation numbers 11, 20, 21, 22 and 25 in the text) and modified the Introduction section to make the discussion of our research gaps and contributions more convincing. The underlined parts below are partial modifications to the Introduction section:

*‘Currently, the common battery capacity optimization methods are generally based on economic indicators [10, 11], technical indicators [12], or a combination of these two indicators [13]... Meanwhile, technical indicators usually consider the battery's voltage and frequency regulation capabilities [13], but there is a lack of technical indicators to optimize battery capacity based on the inherent relationship between PV (photovoltaic) generation and building energy consumption. Since achieving carbon neutrality requires at least 80% renewable energy share [19], increasing the penetration rate of renewable energy is crucial to achieving carbon neutrality. In addition, the problem of PV generation ageing and battery capacity degradation will also cause uncertainty in battery capacity sizing [18]. When selecting the sizing of a solar PV-battery system, references [20-22] consider both the generation and residential sides along with the corresponding technical and economic indicators. However, it lacks consideration for the dynamic degradation of PV generation and battery capacity, which can cause significant errors in optimizing the system size. Up until now, there is a lack of appropriate method for sizing battery capacity in distributed energy systems, and inappropriate methods can result in the undersizing or oversizing of battery capacity, material waste and economic-environmental infeasibility...Centralized systems can achieve better large-scale renewable energy development, improve energy utilization efficiency, and reduce the volatility and uncertainty of the power grid [24, 25] ...’ (Section **Introduction**)*

For specific modifications, please refer to the file ‘Manuscript_R1 with editing marks.docx’.

In addition, we have re-summarized the research questions and research contributions of this article to make it clearer. The research questions are as follows:

‘(1) Considering the carbon intensity of batteries in renewable systems, optimizing battery capacity is crucial to avoiding material waste and economic infeasibility. However, current methods for battery capacity sizing have disadvantages (e.g., most are based on economic indicators and do not reveal the fundamental physical mechanisms of dynamic interconnection and power interaction within integrated photovoltaic-battery-consumer energy systems), leading to capacity overestimation or underestimation. A few methods based on technical indicators only apply to centralized systems or PV/wind turbine farms, and are not suitable for distributed systems.

(2) Centralized photovoltaic systems focus primarily on providing consistent electricity, while distributed consumer battery systems face the challenge of balancing both on-site renewable energy supply and energy demand simultaneously, making it difficult to optimize battery capacity. In addition, the dynamic degradation of battery capacity and PV power generation will also cause difficulties in battery capacity optimization.

(3) The quantification of lifecycle carbon emissions and carbon intensity for building-photovoltaic-battery integrated systems is complicated due to the incomplete database on embodied carbon emission, process uncertainty and overlap of each integrated component's operating stage. Additionally, there has been no comparison of centralized and distributed energy systems in terms of lifecycle carbon intensity and net present value for sustainable electrification lifecycle transformation. As a result, clear sustainable development guidelines for the future battery circular economy are still needed for planning, designing, and operation.' (Section **Introduction**)

To solve these research questions, this paper makes the following contributions:

'1) Developing a scientific and generic methodology for capacity sizing under the distributed prosumer-battery framework considering demand-side management, PV generation degradation and battery dynamic ageing, which will ensure the techno-economic-environmental performance optimality and avoid capacity undersizing or oversizing and resource waste.

2) Quantifying the lifecycle carbon intensity of an integrated building-photovoltaic-battery system with an overlap in operation stages, together with performance superiority through comparison between centralized and distributed energy systems in lifecycle carbon intensity and net present value.

3) Exploring LCA and zero-carbon transformation of both centralized PV-battery-consumer and distributed prosumer-battery systems in different climate zones, and developing climate-adaptive zero-carbon circular economy with a systematic battery capacity sizing methodology and clear guidelines for sustainability transition in planning, designing and operating.' (Section **Introduction**)

If there is any more comment, please suggest them, and we will be very glad to make further improvements for this paper draft.

References:

[A] Cervone, A., Carbone, G., Santini, E., and Teodori, S. (2016). Optimization of the battery size for PV systems under regulatory rules using a Markov-Chains approach. *Renewable Energy* 85, 657-665. 10.1016/j.renene.2015.07.007.

[10] Comello, S., and Reichelstein, S. (2019). The emergence of cost effective battery storage. *Nat Commun* 10, 2038. 10.1038/s41467-019-09988-z.

[11] Bhamidi, L., and Sivasubramani, S. (2021). Optimal Sizing of Smart Home Renewable Energy Resources and Battery Under Prosumer-Based Energy Management. *IEEE Systems Journal* 15, 105-113. 10.1109/JSYST.2020.2967351.

[12] Zhang, Y., Lundblad, A., Campana, P.E., Benavente, F., and Yan, J. (2017). Battery sizing and rule-based operation of grid-connected photovoltaic-battery system: A case study in Sweden. *Energy Conversion and Management* 133, 249-263. 10.1016/j.enconman.2016.11.060.

- [13] Yang, Y., Bremner, S., Menictas, C., and Kay, M. (2018). Battery energy storage system size determination in renewable energy systems: A review. *Renewable and Sustainable Energy Reviews* 91, 109-125. 10.1016/j.rser.2018.03.047.
- [18] Kumtepe, V., and Howey, D.A. (2022). Understanding battery aging in grid energy storage systems. *Joule* 6, 2250-2252. 10.1016/j.joule.2022.09.014.
- [19] Tong, D., Farnham, D.J., Duan, L., Zhang, Q., Lewis, N.S., Caldeira, K., and Davis, S.J. (2021). Geophysical constraints on the reliability of solar and wind power worldwide. *Nat Commun* 12, 6146. 10.1038/s41467-021-26355-z.
- [20] Shankar, A., Vijayakumar, K., Babu, B.C., and Kaur, R. (2022). Energy Trilemma Index-Based Multiobjective Optimal Sizing of PV-Battery System for a Building in Tropical Savanna Climate. *IEEE Systems Journal* 16, 5630-5638. 10.1109/JSYST.2022.3167166.
- [21] Gabr, A.Z., Helal, A.A., and Abbasy, N.H. (2021). Multiobjective Optimization of Photo Voltaic Battery System Sizing for Grid-Connected Residential Prosumers Under Time-of-Use Tariff Structures. *IEEE Access* 9, 74977-74988. 10.1109/ACCESS.2021.3081395.
- [22] Aghamohamadi, M., Mahmoudi, A., and Haque, M.H. (2021). Two-Stage Robust Sizing and Operation Co-Optimization for Residential PV–Battery Systems Considering the Uncertainty of PV Generation and Load. *IEEE Transactions on Industrial Informatics* 17, 1005-1017. 10.1109/TII.2020.2990682.
- [24] Stennikov, V., Barakhtenko, E., Mayorov, G., Sokolov, D., and Zhou, B. (2022). Coordinated management of centralized and distributed generation in an integrated energy system using a multi-agent approach. *Applied Energy* 309. 10.1016/j.apenergy.2021.118487.
- [25] Kani, S.A.P., Wild, P., and Saha, T.K. (2020). Improving Predictability of Renewable Generation Through Optimal Battery Sizing. *IEEE Transactions on Sustainable Energy* 11, 37-47. 10.1109/TSSTE.2018.2883424.
- [26] Peters, I.M., Breyer, C., Jaffer, S.A., Kurtz, S., Reindl, T., Sinton, R., and Vetter, M. (2021). The role of batteries in meeting the PV terawatt challenge. *Joule* 5, 1353-1370. 10.1016/j.joule.2021.03.023.

(2) It would be beneficial to elaborate on the U-value and M-value approaches utilized in this study for capacity sizing. While the U-value traditionally pertains to assessing a product's insulation properties, its adaptation for this research warrants further justification.

Reply: Thank you very much for your very professional comment. In this revision R1, according to your comments, we add descriptions on U-value and M-value to ensure readers can easily follow. We add the description in the section *Battery capacity sizing—centralized PV-battery-consumer system and distributed prosumer-battery system*. Note that, the U-value here is not thermal transmittance, but **Uniformity Value**, cited from [26] *Peters IM, Breyer C, Jaffer SA, Kurtz S, Reindl T, Sinton R, et al. The role of batteries in meeting the PV terawatt challenge. Joule. 2021;5:1353-70.*

The U-value approach introduced in this article is only suitable for centralized energy systems. In other words, the U-value for battery capacity sizing is only considering the process of renewable battery charging, without considering the process of battery discharging based on energy demand. Therefore, considering both renewable to battery charging, and battery to demands discharging processes, we further develop the M-value approach based on the U-value approach to make it more suitable for battery capacity sizing in distributed systems.

Specifically, the U-value calculation method is as follows:

First, without the influence of the battery, U-value can be obtained by the following Equation:

$$U = \frac{E_{gen}}{P_{max} \times \Delta t} \quad (1)$$

‘where E_{gen} refers to the renewable generation; P_{max} refers to the peak power of renewable generation in one day, and Δt refers to 24 hours in this research as shown in Figure 2 (a).’ (Section Battery capacity sizing—centralized PV-battery-consumer system and distributed prosumer-battery system)

After the application of batteries, U-value is expressed as “U'-value” and is calculated as follows:

$$U' = \frac{E_{output}}{P'_{max} \times \Delta t} \quad (2)$$

‘where E_{output} refers to the electricity output of the centralized PV-battery system; P'_{max} refers to the peak power of the electricity output in one day, and Δt refers to 24 hours (one day) in this research as shown in Figure 2 (a). Note that P_{max} and P'_{max} are different because the battery stores energy during periods of high power generation and discharges during periods of low power generation. E_{gen} and E_{output} are not the same. This is because there is a loss in battery charging and discharging.’ (Section Battery capacity sizing—centralized PV-battery-consumer system and distributed prosumer-battery system)

Later, the annual U'-value of the system is obtained based on the average of the daily U'-values. Note that U'-value is related to battery capacity, a cubic function is used for fitting:

$$y = ax^3 + bx^2 + cx + d \quad (3)$$

We further find its slope:

$$y' = 3ax^2 + 2bx + c \quad (4)$$

‘The optimal battery capacity is identified at the largest slope in the mathematical relationship between U-value (y) and battery capacity (x). This is because that, the U-value reaches a peak as the battery capacity increases at the largest slope, which can achieve better results in energy storage, while the continuous increase in the battery capacity will reduce the U-value increase rate, and the battery can no longer perform its special functions.’ (Section Battery capacity sizing—centralized PV-battery-consumer system and distributed prosumer-battery system)

It is noted that the calculation of the U-value in the above method does not consider energy demand, so we developed the M-value method. The calculation of the M-value method is as follows:

Firstly, the value of M can be determined based on the PV capacity, building energy demand, and battery energy storage. The calculation of M:

$$M_{\text{gen}} = \frac{E_{\text{surp}} + E_{\text{match}}}{E_{\text{surp}} + E_{\text{match}} + E_{\text{short}}} \quad (5)$$

$$M_{\text{dem}} = \frac{E_{\text{short}} + E_{\text{match}}}{E_{\text{surp}} + E_{\text{match}} + E_{\text{short}}} \quad (6)$$

$$M = (M_{\text{gen}} \times M_{\text{dem}})^{1/2} \quad (7)$$

‘where E_{surp} refers to the surplus renewable energy of the distributed PV-battery system; E_{match} refers to the self-consumption energy; E_{short} refers to the demand shortage of the distributed prosumer-battery system’ (Section *Battery capacity sizing—centralized PV-battery-consumer system and distributed prosumer-battery system*)

After incorporating the battery into the system, the M-value changes and is denoted as M'-value. The calculation process of M'-value is as follows:

$$M'_{\text{gen}} = \frac{E_{\text{surp}} + E_{\text{match}}}{E_{\text{surp}} + E_{\text{match}} + E_{\text{short}} - E_{\text{ch}}} \quad (8)$$

$$M'_{\text{dem}} = \frac{E_{\text{short}} + E_{\text{match}}}{E_{\text{surp}} + E_{\text{match}} + E_{\text{short}} - E_{\text{dis}}} \quad (9)$$

$$M' = (M'_{\text{gen}} \times M'_{\text{dem}})^{1/2} \quad (10)$$

‘where E_{ch} and E_{dis} refer to the energy charged and discharged to/from batteries. This method determines the required battery size based on matching the renewable generation curve and the building demand curve. Subsequently, the annual M-value of the system is obtained based on the average of the daily M-values. By using this method, the relationship between the M-value and battery capacities can be obtained as shown in Figure 2 (d).’ (Section *Battery capacity sizing—centralized PV-battery-consumer system and distributed prosumer-battery system*)

It is noted that the results of M'-value vary at different battery capacities, and exhibit an S-shaped curve relationship with battery capacity. Therefore, this paper fits the relationship using the following equation:

$$y = \frac{a}{1 + be^{-cx}} \quad (11)$$

Finally, through the NPV results, we found: ‘it is noted that the battery capacity represented by 1/2 of the maximum slope is very close to the battery capacity that maximizes the system NPV. Therefore, it can be considered that the battery capacity represented by 1/2 of the maximum slope in the mathematical relationship between M-value (y) and battery capacity (x) is the optimal battery capacity.’ (Section *Battery capacity sizing—centralized PV-battery-consumer system and distributed prosumer-battery system*)

Figure 2 The battery capacity sizing methods. (a) U-value calculation for centralized PV-battery-consumer system; (b) M-value calculation for distributed PV-battery-building system; (c) battery capacity by U-value approach in the centralized PV-battery-consumer system; (d) battery capacity by M-value approach in the distributed PV-battery-building system.

If there are any more comments, please suggest them, and we will be very glad to make further improvements for this paper draft.

(3) I suggest including a graphical representation to demonstrate how the proposed Circular Economy Framework was applied in this study for clarity and better understanding.

Reply: Thank you very much for your very professional comment. Based on your suggestion, we added a new section ‘The lifecycle design for the renewable-battery-prosumer system’ with Figure 1, where Figure 1 (e) describes the circular economy framework.

Figure 1 Lifecycle design and approach for energy systems (a) Basic information for the system, including weather data, electricity price, renewable resource distribution and grid power source; (b) Designs of traditional energy systems, centralized PV-battery-consumer systems and distributed prosumer-battery systems; (c) Optimal system sizing approaches for centralized and distributed system; (d) research questions of the study; (e) Lifecycle assessment and design guidelines for integrative energy systems.

‘...while Figure 1 (e) further illustrates the application of the circular economy framework in the building energy system. The building lifecycle is categorized into four stages: raw materials, transportation and construction, operation, and demolition and reuse. Likewise, the photovoltaic lifecycle is classified into four stages: raw materials and processing, production, operation, and demolition and recycling. Similarly, the battery's lifecycle is divided into four stages: materials mining, manufacturing, operation, and recycling. It is important to note that these stages overlap during the operation stage. Therefore, it is necessary to differentiate between embodied carbon and operational carbon for the LCA of the entire energy system. Furthermore, the lifecycle carbon intensity chart in Figure 1 (e) shows the carbon emission changes from traditional systems to distributed energy systems for a hotel building. The total carbon intensity of the traditional system is as high as 2021.5 kg CO_{2,e}/m² (including 350.71 kg CO_{2,e}/m² of building embodied carbon intensity and 1670.79 kg CO_{2,e}/m² of operational carbon intensity). After designing the system by the distributed building-PV-prosumer guidelines proposed in this study, the total

carbon intensity of the system decreased to 1121.84 kg CO_{2,e}/m². While the embodied carbon intensities of the building, PVs, and batteries are 350.71, 106.72, and 74.20 kg CO_{2,e}/m², respectively, and the system's operational carbon emissions are 590.21 kg CO_{2,e}/m². This demonstrates the effectiveness of the distributed building-PV-prosumer design guidelines proposed in this paper.' (Section The lifecycle design for the renewable-battery-prosumer system)

If there are any more comments, please suggest them, and we will be very glad to make further improvements for this paper draft.

Reviewer #2 (Remarks to the Author): The paper presents a design approach for integrated PV-battery-consumer energy systems aimed at battery capacity sizing. Lifecycle carbon intensity for the integrated system is quantified, and a design guideline based on area-renewable capacity-battery capacity is provided. While the paper has some merits, its contributions are marginal. The structure of the paper is fragmented, which makes it difficult to follow the story. More comments as follows:

Reply: Thank you very much for your precious time and professional comment. According to your suggestion, we have made detailed revisions to clarify the contributions of this paper and improve the structure to enhance its conciseness and comprehensibility. Descriptions on modifications are summarised below, and more detailed revisions are shown in following-up point-to-point answers.

1. In our response to question 1, we explained our research motivation and re-summarized the research questions addressed and contributions made in this paper.
2. We have added Figure 1 at the beginning of the Results and Discussion section to provide a detailed explanation of the research framework in this paper. Please refer to our response to question 7 for further details.
3. We reviewed the overall content of the paper once again in accordance with the 'Abstract-Introduction-Results-Discussion-Methods-References' format from the journal submitting guideline. The specific modifications can be found in the document 'Manuscript_R1 with editing marks.docx'.

If there is any more comment, please suggest them, and we will be very glad to make further improvements for this paper draft.

1. The research motivations are not clear. The authors discuss the battery sizing capacity as the main motivation. However, it is not clear what are the research questions that haven't been addressed in the literature.

Reply: Thank you very much for your professional comment. The research motivation for this article is due to our attention to the following facts:

1. Currently, most of the battery sizing methods are based on the PV generation side. For example, the methods mentioned in Ref. [12, 13, 26] only consider the PV generation side, which is not conducive to sizing the battery capacity in distributed energy systems. This is because batteries in distributed energy systems need to balance on-site renewable energy generation and energy demand simultaneously. If the method lacks consideration for energy demand, it will result in deviation in battery capacity sizing,

leading to resource wastage, reduced revenue, and environmental pollution. Therefore, a method that considers both generation and consumption sides (such as buildings and EVs) is essential.

2. Some methods that consider both generation and consumption sides introduce economic and environmental factors, which are not very reliable due to the uncertainty in prices and carbon emissions. For example, Ref. [10] and [A] only consider economic and environmental factors. Only Considering economic and environmental indicators may result in unreasonable battery capacity sizing, as net present value (NPV) and carbon emissions are highly dependent on various independent variables (such as renewable energy costs, battery prices, electricity prices, carbon emission factors, and embodied carbon emissions) and various intermediate variables (such as renewable penetration rates, demand shortages, battery charge and discharge power, and battery degradation rates). Therefore, the current battery sizing will be underestimated or overestimated in the future due to price fluctuations and improvements in technology levels (which decrease carbon emissions). Moreover, battery capacity sizing based on economic and environmental indicators cannot reveal the basic physical mechanisms of dynamic interconnection and power interaction within the integrated PV-battery-consumer energy system. Therefore, a battery capacity sizing method that can satisfy the dynamic balance between the generation and consumption sides is necessary.

3. Currently, there is a lack of analysis on the impact of capacity sizing methods on the life cycle carbon footprint when comprehensively considering both embodied carbon in raw materials of renewable systems and batteries, and operational carbon with their dynamic performance degradation and replacement times.

Based on the above facts, we summarized the main research questions as follows:

‘(1) Considering the carbon intensity of batteries in renewable systems, optimizing battery capacity is crucial to avoiding material waste and economic infeasibility. However, current methods for battery capacity sizing have disadvantages (e.g., most are based on economic indicators and do not reveal the fundamental physical mechanisms of dynamic interconnection and power interaction within integrated photovoltaic-battery-consumer energy systems), leading to the capacity overestimation or underestimation. A few methods based on technical indicators only apply to centralized systems or PV/wind turbine farms, and are not suitable for distributed systems.

(2) Centralized photovoltaic systems focus primarily on providing consistent electricity, while distributed consumer battery systems face the challenge of balancing both on-site renewable energy supply and energy demand simultaneously, making it difficult to optimize battery capacity. In addition, the dynamic degradation of battery capacity and PV power generation will also cause difficulties in battery capacity optimization.

(3) The quantification of lifecycle carbon emissions and carbon intensity for building-photovoltaic-battery integrated systems is complicated due to the overlap of each integrated component's operating stage and the separation of other stages. Additionally, there has been no comparison of centralized and distributed energy systems in terms of lifecycle carbon intensity and net present value for sustainable electrification lifecycle transformation. As a result, clear sustainable development guidelines for the

future battery circular economy are still needed for planning, designing, and operation.' (Section **Introduction**)

The research gaps identified above have led to the following originality and contributions of our research:

'1) Developing a scientific and generic methodology for capacity sizing under the distributed prosumer-battery framework considering demand-side management, PV generation degradation and battery dynamic ageing, which will ensure the techno-economic-environmental performance optimality and avoid capacity undersizing or oversizing and resource waste.

2) Quantifying the lifecycle carbon intensity of an integrated building-photovoltaic-battery system with an overlap in operation stages, together with performance superiority through comparison between centralized and distributed energy systems in lifecycle carbon intensity and net present value.

3) Exploring LCA and zero-carbon transformation of both centralized PV-battery-consumer and distributed prosumer-battery systems in different climate zones, and developing climate-adaptive zero-carbon circular economy with a systematic battery capacity sizing methodology and clear guidelines for sustainability transition in planning, designing and operating.' (Section **Introduction**)

The re-summarized research gaps and contributions have been included in the manuscript, and you can review them in the 'Manuscript_R1 with editing marks.docx' document.

If there is any more comment, please suggest them, and we will be very glad to make further improvements for this paper draft.

References:

[A] Cervone, A., Carbone, G., Santini, E., and Teodori, S. (2016). Optimization of the battery size for PV systems under regulatory rules using a Markov-Chains approach. *Renewable Energy* 85, 657-665. 10.1016/j.renene.2015.07.007.

[10] Comello, S., and Reichelstein, S. (2019). The emergence of cost effective battery storage. *Nat Commun* 10, 2038. 10.1038/s41467-019-09988-z.

[12] Zhang, Y., Lundblad, A., Campana, P.E., Benavente, F., and Yan, J. (2017). Battery sizing and rule-based operation of grid-connected photovoltaic-battery system: A case study in Sweden. *Energy Conversion and Management* 133, 249-263. 10.1016/j.enconman.2016.11.060.

[13] Yang, Y., Bremner, S., Menictas, C., and Kay, M. (2018). Battery energy storage system size determination in renewable energy systems: A review. *Renewable and Sustainable Energy Reviews* 91, 109-125. 10.1016/j.rser.2018.03.047.

[26] Peters, I.M., Breyer, C., Jaffer, S.A., Kurtz, S., Reindl, T., Sinton, R., and Vetter, M. (2021). The role of batteries in meeting the PV terawatt challenge. *Joule* 5, 1353-1370. 10.1016/j.joule.2021.03.023.

2. What does the statement “there is currently a lack of research on the renewable energy penetration rate” mean, and how is it related to this research?

Reply: Thank you very much for your very professional comment. We are sorry that this sentence is not clear. Here is our modification (underlined are specific modifications):

*‘Meanwhile, technical indicators usually consider the battery's voltage and frequency regulation capabilities [13], but there is a lack of technical indicators to optimize battery capacity based on the inherent relationship between PV (photovoltaic) generation and building energy consumption.’ (Section **Introduction**)*

This modification is made because we notice that much research, such as Ref. [B, C, D], only optimize battery capacity from the voltage and frequency regulation capabilities of the battery. The dynamic balancing ability of batteries to the intrinsic relationship between PV power generation and building energy consumption has not been fully considered when optimizing battery capacity.

If there is any more comment, please suggest them, and we will be very glad to make further improvements for this paper draft.

References:

[B] Jia, H., Mu, Y., and Qi, Y. (2014). A statistical model to determine the capacity of battery–supercapacitor hybrid energy storage system in autonomous microgrid. *International Journal of Electrical Power & Energy Systems* 54, 516-524. 10.1016/j.ijepes.2013.07.025.

[C] Yue, M., and Wang, X. (2015). Grid Inertial Response-Based Probabilistic Determination of Energy Storage System Capacity Under High Solar Penetration. *IEEE Transactions on Sustainable Energy* 6, 1039-1049. 10.1109/tste.2014.2328298.

[D] Johnston, L., Díaz-González, F., Gomis-Bellmunt, O., Corchero-García, C., and Cruz-Zambrano, M. (2015). Methodology for the economic optimisation of energy storage systems for frequency support in wind power plants. *Applied Energy* 137, 660-669. 10.1016/j.apenergy.2014.09.031.

[13] Yang, Y., Bremner, S., Menictas, C., and Kay, M. (2018). Battery energy storage system size determination in renewable energy systems: A review. *Renewable and Sustainable Energy Reviews* 91, 109-125. 10.1016/j.rser.2018.03.047.

3. The difference between centralized PV-battery-consumer and distributed systems is not well explained. If the authors are referring to residential PV-battery systems, how can it be considered a centralized system?

Reply: Thank you very much for your very professional comment. To make it more clear, we add a section ‘*The lifecycle design for the renewable-battery-prosumer system*’ to further clarify the difference.

For the centralized PV-battery-consumer system, where “PV” refers to a solar PV farm, the battery is used to regulate the power output from the farm, and the consumer is a high-rise building that consumes electricity. Since this system does not involve on-site renewable systems and their on-site interaction with building energy consumption, it is considered a centralized system.

In order to make it easier for readers to understand, we have added Figure 1 to the Results and Discussion section, which shows the configuration of centralized and distributed systems:

Figure 1 Lifecycle design and approach for energy systems (a) Basic information for the system, including weather data, electricity price, renewable resource distribution and grid power source; (b) Designs of traditional energy systems, centralized PV-battery-consumer systems and distributed prosumer-battery systems; (c) Optimal system sizing approaches for centralized and distributed system; (d) research questions of the study; (e) Lifecycle assessment and design guidelines for integrative energy systems.

‘...Figure 1 (b) reflects the system design of this study, which includes the following three systems: Traditional energy systems do not install any PV panels, but instead rely on thermal power generation for electricity supply, resulting in a high electricity carbon emission factor (about 0.8–1.0 kg CO₂/kWh in different areas) [7]. In addition, there exists an imbalance between power generation and demand in traditional systems, which is addressed by regulating the frequency of thermal power generators to achieve power control. With the gradual transition from traditional energy systems to renewable energy systems, two potential future energy systems have emerged: centralized PV-battery-consumer systems and distributed prosumer-battery systems.

Centralized PV-battery-consumer systems use solar energy to cover the building energy demand. However, the power generation characteristics of PVs are not as stable as traditional thermal power plants, and they fluctuate significantly over time. Therefore, the design idea of centralized PV-battery-consumer energy systems is to concentrate all the energy generated by PVs and supply electricity in a similar way to traditional thermal power plants by energy storage systems. The limitation of this approach is that the energy storage is not set up with building energy demand, which will cause the over- or under-estimate of the battery capacity.

*In distributed prosumer-battery systems, the design idea is that it treats each building-PV-battery energy system as an independent system and can still interact with the external power grid as a whole system. Therefore, this system can fully consider the spatiotemporal mismatching between solar energy and building demand when designing the energy storage system, and better configure the required energy storage battery capacity...’ (Section *The lifecycle design for the renewable-battery-prosumer system*)*

In addition, as mentioned in ref [E], ‘PV systems are divided into two categories in terms of their configuration, namely centralized and distributed. In terms of their connectivity.....Centralized PV systems exist as large solar farms as opposed to distributed PV systems that are installed at or near an individual building Furthermore, off-site installations are usually grid-connected and centralized while on-site installations can be centralized or distributed and standalone or grid-connected.’

Therefore, the main difference between the centralized system and the distributed system in this paper lies in the different PV system configurations. The PV system used in the centralized system of this paper is a PV farm, which collects solar energy from suburban areas and delivers it to buildings. The battery is primarily used to maintain a stable power supply from the PV farm. In contrast, the distributed system in this paper includes three types of PVs: BIPVs, roof-top PVs, and other on-site PVs close to buildings. These PVs generate electricity which is consumed by the buildings on-site. The battery is used to store excess renewable energy and supply electricity when PV generation is insufficient.

If there is any more comment, please suggest them, and we will be very glad to make further improvements for this paper draft.

References:

[7] Song, A., and Zhou, Y. (2023). Advanced cycling ageing-driven circular economy with E-mobility-based energy sharing and lithium battery cascade utilisation in a district community. *Journal of Cleaner Production* 415. 10.1016/j.jclepro.2023.137797.

[26] Peters, I.M., Breyer, C., Jaffer, S.A., Kurtz, S., Reindl, T., Sinton, R., and Vetter, M. (2021). The role of batteries in meeting the PV terawatt challenge. *Joule* 5, 1353-1370. 10.1016/j.joule.2021.03.023.

[E] Aghamolaei, R., Shamsi, M.H., and O'Donnell, J. (2020). Feasibility analysis of community-based PV systems for residential districts: A comparison of on-site centralized and distributed PV installations. *Renewable Energy* 157, 793-808. 10.1016/j.renene.2020.05.024.

4. The contributions of the work are marginal. The paper is more focused on doing an analysis of battery sizing and impact of different metrics rather than proposing a new methodology which improves the existing methods in the literature.

Reply: Thank you very much for your very professional comment. Based on your comments, we summarized the contribution below:

Based on above research gaps, research originality and contributions are listed below:

- 1) Developing a scientific and generic methodology for capacity sizing under the distributed prosumer-battery framework considering demand-side management, PV generation degradation and battery dynamic ageing, which will ensure the techno-economic-environmental performance optimality and avoid capacity undersizing or oversizing and resource waste.
- 2) Quantifying the lifecycle carbon intensity of an integrated building-photovoltaic-battery system with an overlap in operation stages, together with performance superiority through comparison between centralized and distributed energy systems in lifecycle carbon intensity and net present value.
- 3) Exploring LCA and zero-carbon transformation of both centralized PV-battery-consumer and distributed prosumer-battery systems in different climate zones, and developing climate-adaptive zero-carbon circular economy with a systematic battery capacity sizing methodology and clear guidelines for sustainability transition in planning, designing and operating.

Specifically, in this article, we use a method called U-value to size battery capacity. This method comes from [26] *Peters IM, Breyer C, Jaffer SA, Kurtz S, Reindl T, Sinton R, et al. The role of batteries in meeting the PV terawatt challenge. Joule. 2021;5:1353-70.*

However, the U-value method only measures battery capacity based on the PV generation side, making it suitable only for centralized systems. To make it applicable to distributed systems, we further developed the M-value method. The M-value method is compared with the U-value method with high superiorities in sizing battery capacity within distributed energy systems. The M-value method considers both the PV generation side and the energy consumption side, making it more suitable for optimizing battery capacity in distributed energy systems.

Specifically, the calculation of the M-value method is as follows:

Firstly, the value of M can be determined based on the PV capacity, building energy demand, and battery energy storage. The calculation of M:

$$M_{\text{gen}} = \frac{E_{\text{surp}} + E_{\text{match}}}{E_{\text{surp}} + E_{\text{match}} + E_{\text{short}}} \quad (5)$$

$$M_{\text{dem}} = \frac{E_{\text{short}} + E_{\text{match}}}{E_{\text{surp}} + E_{\text{match}} + E_{\text{short}}} \quad (6)$$

$$M = (M_{\text{gen}} \times M_{\text{dem}})^{1/2} \quad (7)$$

‘where E_{surp} refers to the surplus renewable energy of the distributed PV-battery system; E_{match} refers to the self-consumption energy; E_{short} refers to the demand shortage of the distributed prosumer-battery

system' (Section *Battery capacity sizing—centralized PV-battery-consumer system and distributed prosumer-battery system*)

After incorporating the battery into the system, the M-value changes and is denoted as M'-value. The calculation process of M'-value is as follows:

$$M'_{\text{gen}} = \frac{E_{\text{surp}} + E_{\text{match}}}{E_{\text{surp}} + E_{\text{match}} + E_{\text{short}} - E_{\text{ch}}} \quad (8)$$

$$M'_{\text{dem}} = \frac{E_{\text{short}} + E_{\text{match}}}{E_{\text{surp}} + E_{\text{match}} + E_{\text{short}} - E_{\text{dis}}} \quad (9)$$

$$M' = (M'_{\text{gen}} \times M'_{\text{dem}})^{1/2} \quad (10)$$

'where E_{ch} and E_{dis} refer to the energy charged and discharged to/from batteries. This method determines the required battery size based on matching the renewable generation curve and the building demand curve. Subsequently, the annual M-value of the system is obtained based on the average of the daily M-values. By using this method, the relationship between the M-value and battery capacities can be obtained as shown in Figure 2 (d).' (Section *Battery capacity sizing—centralized PV-battery-consumer system and distributed prosumer-battery system*)

It is noted that the results of M'-value vary at different battery capacities, and exhibit an S-shaped curve relationship with battery capacity. Therefore, this paper fits the relationship using the following equation:

$$y = \frac{a}{1 + be^{-cx}} \quad (11)$$

Finally, through the NPV results, we found: 'it is noted that the battery capacity represented by 1/2 of the maximum slope is very close to the battery capacity that maximizes the system NPV. Therefore, it can be considered that the battery capacity represented by 1/2 of the maximum slope in the mathematical relationship between M-value (y) and battery capacity (x) is the optimal battery capacity.' (Section *Battery capacity sizing—centralized PV-battery-consumer system and distributed prosumer-battery system*)

Figure 2 The battery capacity sizing methods. (a) U-value calculation for centralized PV-battery-consumer system; (b) M-value calculation for distributed PV-battery-building system; (c) battery capacity by U-value approach in the centralized PV-battery-consumer system; (d) battery capacity by M-value approach in the distributed PV-battery-building system.

If there are any more comments, please suggest them, and we will be very glad to make further improvements for this paper draft.

5. Why are the results and discussion presented right after the introduction before explaining the methodology?

Reply: Thank you very much for your very professional comment. Strictly following the submission guidelines of the journal: <https://www.nature.com/documents/commsj-phys-style-formatting-guide-accept.pdf>, we have to place the results and discussion directly after the introduction, forming 'Abstract-Introduction-Results-Discussion-Methods-References'. Moreover, we agree that putting Methods

before the results can be more helpful to understand, but the Methods should be put at the end according to the submission guidelines of the journal.

If there are any more comments, please suggest them, and we will be very glad to make further improvements for this paper draft.

6. Equations should be explained in the main body not in the figures.

Reply: Thank you very much for your very professional comment. We have deleted equations in Figure 1 (Now it's Figure 2) and placed them in the text for explanation. The revised figure and text are as follows:

Figure 2 The battery capacity sizing methods. (a) U-value calculation for centralized PV-battery-consumer system; (b) M-value calculation for distributed PV-battery-building system; (c) battery capacity by U-value approach in the centralized PV-battery-consumer system; (d) battery capacity by M-value approach in the distributed PV-battery-building system.

‘Centralized PV-battery-consumer system: Capacity sizing on centralized PV-battery-consumer systems includes two aspects, i.e., the PV capacity sizing and battery capacity sizing. In terms of centralized PV-battery-consumer systems, the PV capacity sizing is based on the zero-energy principle, i.e., the electricity production of PVs needs to meet the building energy consumption in total. However, the battery capacity sizing is a little complicated. In order to determine the battery capacity of a centralized PV-battery system, the concept of uniformity is applied [26]. Specifically, for a PV system without batteries, its uniformity (U-value) can be calculated according to the Equation (1) as shown in Figure 2 (a):

$$U = \frac{E_{gen}}{P_{max} \times \Delta t} \quad (1)$$

where E_{gen} refers to the renewable generation; P_{max} refers to the peak power of renewable generation in one day, and Δt refers to 24 hours in this research as shown in Figure 2 (a).

After the battery is deployed, the energy output curve of the centralized PV-battery system will change as shown in Figure 2 (b). Therefore, the changed U-value (U') will be calculated according to the Equation (2):

$$U' = \frac{E_{output}}{P'_{max} \times \Delta t} \quad (2)$$

where E_{output} refers to the electricity output of the centralized PV-battery system; P'_{max} refers to the peak power of the electricity output in one day, and Δt refers to 24 hours (one day) in this research as shown in Figure 2 (a). Note that P_{max} and P'_{max} are different because the battery stores energy during periods of high power generation and discharges during periods of low power generation. E_{gen} and E_{output} are not the same. This is because there is a loss in battery charging and discharging.

Subsequently, the annual U-value of the system is obtained based on the average of the daily U-values. Based on the method shown in Equations (1-2), the relationship between U-value and battery capacity is obtained, as shown in Figure 2 (c), where the relationship between U-value (y) and battery capacity (x) is fitted by the Equation (3):

$$y = ax^3 + bx^2 + cx + d \quad (3)$$

The slope of the equation can also be found in Equation (4):

$$y' = 3ax^2 + 2bx + c \quad (4)$$

The optimal battery capacity is identified at the largest slope in the mathematical relationship between U-value (y) and battery capacity (x). This is because that, the U-value reaches a peak as the battery capacity increases at the largest slope, which can achieve better results in energy storage, while the continuous increase in the battery capacity will reduce the U-value increase rate, and the battery can no longer perform its special functions.

Distributed prosumer-battery system: Considering the difference between the distributed PV-battery system and the centralized system, in addition to allowing the PV to supply stable electric energy, the distributed system also needs to consider the matching of the renewable generation and the electricity demand spatiotemporally. Therefore, we define a matching degree (M-value) calculated as shown in the Figure 2 (b) and Equations (5-7):

$$M_{gen} = \frac{E_{surp} + E_{match}}{E_{surp} + E_{match} + E_{short}} \quad (5)$$

$$M_{dem} = \frac{E_{short} + E_{match}}{E_{surp} + E_{match} + E_{short}} \quad (6)$$

$$M = (M_{gen} \times M_{dem})^{1/2} \quad (7)$$

where E_{surp} refers to the surplus renewable energy of the distributed PV-battery system; E_{match} refers to the self-consumption energy; E_{short} refers to the demand shortage of the distributed prosumer-battery system.

When the battery is deployed, the matching degree (M-value) will be updated as shown in Equations (8-10):

$$M'_{gen} = \frac{E_{surp} + E_{match}}{E_{surp} + E_{match} + E_{short} - E_{ch}} \quad (8)$$

$$M'_{dem} = \frac{E_{short} + E_{match}}{E_{surp} + E_{match} + E_{short} - E_{dis}} \quad (9)$$

$$M' = (M'_{gen} \times M'_{dem})^{1/2} \quad (10)$$

where E_{ch} and E_{dis} refer to the energy charged and discharged to/from batteries. This method determines the required battery size based on matching the renewable generation curve and the building demand curve. Subsequently, the annual M-value of the system is obtained based on the average of the daily M-values. By using this method, the relationship between the M-value and battery capacities can be obtained as shown in Figure 2 (d).

Note that the relationship between battery capacity and M-value shows the pattern of the S-curve in Figure 2 (d), so they are fitted through the S-curve fitting equation as shown in Equation (11):

$$y = \frac{a}{1 + be^{-cx}} \quad (11)$$

According to the NPV results in Figure 2 (d), it is noted that the battery capacity represented by 1/2 of the maximum slope is very close to the battery capacity that maximizes the system NPV. Therefore, it can be considered that the battery capacity represented by 1/2 of the maximum slope in the mathematical relationship between M-value (y) and battery capacity (x) is the optimal battery capacity.' (Section Battery capacity sizing—centralized PV-battery-consumer system and distributed prosumer-battery system)

If there are any more comments, please suggest them, and we will be very glad to make further improvements for this paper draft.

References:

[26] Peters, I.M., Breyer, C., Jaffer, S.A., Kurtz, S., Reindl, T., Sinton, R., and Vetter, M. (2021). The role of batteries in meeting the PV terawatt challenge. *Joule* 5, 1353-1370. 10.1016/j.joule.2021.03.023.

7. The paper needs to be restructured. It is difficult to follow different sections.

Reply: Thank you very much for your very professional comment. In response to your concerns and requirements, the paper has been restructured and the following adjustments have been made:

1. We reviewed the overall content of the paper once again in accordance with the 'Abstract-Introduction-Results-Discussion-Methods-References' format from the journal submitting guideline. The specific modifications can be found in the document 'Manuscript_R1 with editing marks.docx'.

In the Abstract section, we described the research background, gaps, contributions, results, and significance of the paper.

In the Introduction section, we conducted a literature review of previous studies and elaborated on the current research gaps, research contributions, and research content of this paper.

In the Results section, we presented the energy system designed in this paper, the battery capacity sizing processes achieved through the U-value and M-value methods, the life cycle carbon intensity calculation of the building-PV-battery system based on the circular economy framework, and the climate adaptability of this method.

In the Discussion section, we summarized the main viewpoints and conclusions of the paper and proposed possible future research directions.

In the Methods section, we summarized the methods used in this paper, including the circular economy framework and the Li-ion battery degradation model, as well as the evaluation criteria used in this paper, including NPV and carbon emissions/carbon footprint.

Therefore, this paper starts with the research gaps shown in the Introduction section, and uses various models and evaluation criteria. By inputting external information such as weather, energy distribution in the grid, and carbon emission factors into the models, the Results section presents the battery capacity adjustment process achieved by the U-value and M-value methods, as well as the life cycle carbon intensity of the building-PV-battery system based on the circular economy framework. Finally, the Discussion section summarizes the main findings and limitations of the paper.

2. We have added Figure 1 to make it easier for readers to understand the topic and structure of the article. The specific modifications are as follows:

Figure 1 Lifecycle design and approach for energy systems (a) Basic information for the system, including weather data, electricity price, renewable resource distribution and grid power source; (b) Designs of traditional energy systems, centralized PV-battery-consumer systems and distributed prosumer-battery systems; (c) Optimal system sizing approaches for centralized and distributed system; (d) research questions of the study; (e) Lifecycle assessment and design guidelines for renewable energy systems.

‘Figure 1 illustrates the model development, system establishment, and circular economy framework of this study. As shown in Figure 1 (a), the basic information required for this study is shown in a schematic form, including meteorological parameters, electricity price data, renewable energy distribution, and power sources of the grid. Figure 1 (b) reflects the system design of this study, which includes the following three systems:

Traditional energy systems do not install any PV panels, but instead rely on thermal power generation for electricity supply, resulting in a high electricity carbon emission factor (about 0.8~1.0 kg CO₂/kWh in different areas) [7]. In addition, there exists an imbalance between power generation and demand in traditional systems, which is addressed by regulating the frequency of thermal power generators to achieve power control. With the gradual transition from traditional energy systems to renewable energy

systems, two potential future energy systems have emerged: centralized PV-battery-consumer systems and distributed prosumer-battery systems.

Centralized PV-battery-consumer systems use solar energy to cover the building energy demand. However, the power generation characteristics of PVs are not as stable as traditional thermal power plants, and they fluctuate significantly over time. Therefore, the design idea of centralized PV-battery-consumer energy systems is to concentrate all the energy generated by PVs and supply electricity in a similar way to traditional thermal power plants by energy storage systems. The limitation of this approach is that the energy storage is not set up with building energy demand, which will cause the over- or under-estimate of the battery capacity.

In distributed prosumer-battery systems, the design idea is that it treats each building-PV-battery energy system as an independent system and can still interact with the external power grid as a whole system. Therefore, this system can fully consider the spatiotemporal mismatching between solar energy and building demand when designing the energy storage system, and better configure the required energy storage battery capacity.

Figure 1 (c) illustrates the design approach for the PV and battery capacity in the system. In traditional systems, electricity generation can be adjusted by regulating the generator frequency, but this is not possible in renewable energy systems. Therefore, in this study, a centralized renewable energy system is developed to supply stable power through energy storage systems, and the battery capacity is optimized using the U-value method [26]. In distributed energy systems, after determining the required PV capacity based on the net-zero energy principle, battery capacity optimization is achieved by using the M-value method developed in this study. Figure 1 (d) shows the five most important research questions addressed in this study, while Figure 1 (e) further illustrates the application of the circular economy framework in the building energy system. The building lifecycle is categorized into four stages: raw materials, transportation and construction, operation, and demolition and reuse. Likewise, the photovoltaic lifecycle is classified into four stages: raw materials and processing, production, operation, and demolition and recycling. Similarly, the battery's lifecycle is divided into four stages: materials mining, manufacturing, operation, and recycling. It is important to note that these stages overlap during the operation stage. Therefore, it is necessary to differentiate between embodied carbon and operational carbon for the LCA of the entire energy system. Furthermore, the lifecycle carbon intensity chart in Figure 1 (e) shows the carbon emission changes from traditional systems to distributed energy systems for a hotel building. The total carbon intensity of the traditional system is as high as 2021.5 kg CO₂/m² (including 350.71 kg CO₂/m² of building embodied carbon intensity and 1670.79 kg CO₂/m² of operational carbon intensity). After designing the system by the distributed building-PV-prosumer guidelines proposed in this study, the total carbon intensity of the system decreased to 1121.84 kg CO₂/m². While the embodied carbon intensities of the building, PVs, and batteries are 350.71, 106.72, and 74.20 kg CO₂/m², respectively, and the system's operational carbon emissions are 590.21 kg CO₂/m². This demonstrates the effectiveness of the distributed building-PV-prosumer design guidelines proposed in this paper. (Section The lifecycle design for the renewable-battery-prosumer system)

If there are any more comments, please suggest them, and we will be very glad to make further improvements for this paper draft.

References:

[7] Song, A., and Zhou, Y. (2023). Advanced cycling ageing-driven circular economy with E-mobility-based energy sharing and lithium battery cascade utilisation in a district community. *Journal of Cleaner Production* 415. 10.1016/j.jclepro.2023.137797.

[26] Peters, I.M., Breyer, C., Jaffer, S.A., Kurtz, S., Reindl, T., Sinton, R., and Vetter, M. (2021). The role of batteries in meeting the PV terawatt challenge. *Joule* 5, 1353-1370. 10.1016/j.joule.2021.03.023.

Reviewer #3 (Remarks to the Author):

Review report

This study focuses on carbon neutrality, instead of technical and economic aspects, of the PV-BESS-Prosumer set. It aims to develop a methodology for integrated system capacity sizing with lifecycle carbon intensity quantification and net present value for techno-economic environmental performance optimality. The study proposes low-carbon design guidelines for integrated PV-battery-consumer energy systems with climate adaptability, to assist decision making among planners, designers, managers and policy-makers.

Reply: Thank you very much for taking valuable time to help review our work and your very professional comments. Our study proposes low-carbon design guidelines for PV-battery-consumer energy systems with climate adaptability, which we believe can assist decision-making among planners, designers, managers, and policymakers. We will revise the manuscript to further emphasize the importance of considering technical and economic aspects in the design of the PV-battery-consumer energy systems.

The subject is timely, the proposed idea is practically interesting and the methodology sounds. However, I have still concerns about clarity of the paper; some of them are:

1. Most of the existing battery sizing optimization techniques consider economic and technical objectives to achieve the best using minimum battery size. They mostly consider a constraint on battery depth of charge and also on overcharging; thus, battery degradation is also considered. My understanding is that using those methods, with minimum BESS size and low degradation, the best option in terms of carbon neutrality in production and recycling has already been achieved (although it may not be directly claimed). Therefore, I have a question about the real contribution of the paper. Authors can clarify their contribution by comparing the results of their method with one recent literature with technical or economic objectives.

Reply: Thank you for your valuable comments on our paper. Based on your feedback, we have included a comparison between the method based on maximizing economic and environmental benefits and our study. The following content has been added to the original text:

‘To further illustrate the rationality of the M-value method for battery capacity sizing in distributed systems, the results are compared with Ref 32, which uses multi-objective optimization to determine the

optimal battery capacity for hotel, office, and residential buildings based on maximizing economic and environmental benefits. The optimal battery capacities are found to be 895.6 kWh, 900 kWh and 895.6 kWh (which means a battery capacity intensity of 0.0746, 0.0750 and 0.0746 kWh/m² based on an area of 12,000 m²), respectively 32. When these battery capacity intensities are applied to Guangzhou, the results showed an overall lower net present value (NPV) and higher carbon intensity compared to the method used in this study as shown in Figure 6. Specifically, the NPV for the hotel building is 93.70 US\$/m², which is much lower than 127.09 US\$/m² for the results in this study. Furthermore, the carbon intensities for the research method from Ref 32 are 1526.94, 756.37, and 987.75 kg CO_{2,e}/m², respectively, while they are higher than 1121.84, 736.20 and 720.59 kg CO_{2,e}/m² in this study. Therefore, it can be noticed that the method proposed in this study has more advantages than traditional approaches based on maximizing economic and environmental benefits in NPV and carbon emissions.'

Figure 6 Comparison on different battery sizing approaches in NPV and carbon intensity.' (Section Comparison on different battery capacity sizing methods in distributed systems)

If there is any more comment, please suggest them, and we will be very glad to make further improvements for this paper draft.

References:

[32] Zhou, Y., Cao, S., Kosonen, R., and Hamdy, M. (2020). Multi-objective optimisation of an interactive buildings-vehicles energy sharing network with high energy flexibility using the Pareto archive NSGA-II algorithm. Energy Conversion and Management 218. 10.1016/j.enconman.2020.113017.

2. The proposed methodology assumes that all renewable generation must be either consumed or stored in BESS, at all times. Although a reasonable assumption for grid planning, this methodology may cause too conservative results considering that the power grid can tolerate the specific amount of extra

generation based on available standards, e.g. IEC60038. This assumption can reduce the size of required BESS, both in centralised and decentralised cases; thus contributing to carbon neutrality.

Reply: Thank you very much for your very professional comment. The reasons for making such conservative assumptions are mainly as follows:

1. Due to the lack of unified coordination and planning between photovoltaic power generation and power grid construction in China, local power grids are often unable to absorb the excess power generated by photovoltaics. As mentioned in Ref. [41]: *‘On the one hand, the PV power stations in China’s PV resource-rich northwest region access the grid large scale and intensively, but they are not synchronized with the local power grid and lack flexible power-source-adjustment planning. As a result, the locality cannot absorb PV power; it is difficult to transmit power and the “power abandonment” phenomenon is emerging with regard to PV power stations. On the other hand, grid access technology and the relevant policies of high proportion distributed PV systems need to be investigated and planned, while the relevant power distribution network requires unified coordination.’*

2. Although the Chinese government encourages distributed photovoltaics and grid connection, many local governments have been slow to introduce policies, hindering the progress of photovoltaic power grid connection. As stated in Ref. [42]: *‘Government time lags in policy implementation are a crucial barrier to DG PV development. The central government’s DG PV policies proceed slowly through dissemination by provincial and local governments.’*

Therefore, based on the above reasons, we make the conservative assumption that PV is not connected to the grid.

In this Revision R1, to help readers’ to easily follow, we add the description below:

‘Note that the renewable energy system in this study does not sell electricity to the grid. This is due to the lack of unified coordination between China’s photovoltaic power generation and power grid construction⁴¹, as well as the slow introduction of local policies⁴². Therefore, a conservative assumption is made in the study.’ (Section Assessment criteria-EXPERIMENTAL PROCEDURES)

If there is any more comment, please suggest them, and we will be very glad to make further improvements for this paper draft.

References:

[41] Sun, H., Zhi, Q., Wang, Y., Yao, Q., and Su, J. (2014). China’s solar photovoltaic industry development: The status quo, problems and approaches. *Applied Energy* 118, 221-230. 10.1016/j.apenergy.2013.12.032.

[42] Zhang, F., Deng, H., Margolis, R., and Su, J. (2015). Analysis of distributed-generation photovoltaic deployment, installation time and cost, market barriers, and policies in China. *Energy Policy* 81, 43-55. 10.1016/j.enpol.2015.02.010.